# Mettl3-mediated m⁶A modification of Fgf16 restricts cardiomyocyte proliferation during heart regeneration

Fu-Qing Jiang[1†], Kun Liu[1†], Jia-Xuan Chen[1†], Yan Cao[1], Wu-Yun Chen[1], Wan-Ling Zhao[1], Guo-Hua Song[2], Chi-Qian Liang[1], Yi-Min Zhou[1], Huan-Lei Huang[3], Rui-Jin Huang[4], Hui Zhao[5], Kyu-Sang Park[6], Zhenyu Ju[7]*, Dongqing Cai[1]*, Xu-Feng Qi[1]*

[1]Key Laboratory of Regenerative Medicine of Ministry of Education, Department of Developmental & Regenerative Biology, Jinan University, Guangzhou, China; [2]College of Basic Medical Sciences, Shandong First Medical University & Shandong Academy of Medical Science, Tai'an, China; [3]Department of Cardiovascular Surgery, Guangdong Cardiovascular Institute, Guangdong Provincial People's Hospital & Guangdong Academy of Medical Sciences, Guangzhou, China; [4]Department of Neuroanatomy, Institute of Anatomy, University of Bonn, Bonn, Germany; [5]Stem Cell and Regeneration TRP, School of Biomedical Sciences, Chinese University of Hong Kong, Hong Kong, China; [6]Department of Physiology, Wonju College of Medicine, Yonsei University, Wonju, Republic of Korea; [7]Key Laboratory of Regenerative Medicine of Ministry of Education, Institute of Aging and Regenerative Medicine, Guangzhou, China

**\*For correspondence:**
zhenyuju@163.com (ZJ);
tdongbme@jnu.edu.cn (DC);
qixufeng@jnu.edu.cn (X-FQ)

†These authors contributed
equally to this work

**Competing interest:** The authors
declare that no competing
interests exist.

**Reviewing Editor:** Hina W
Chaudhry, Icahn School of
Medicine at Mount Sinai, United
States

**Abstract** Cardiovascular disease is the leading cause of death worldwide due to the inability of adult heart to regenerate after injury. $N^6$-methyladenosine (m⁶A) methylation catalyzed by the enzyme methyltransferase-like 3 (Mettl3) plays an important role in various physiological and pathological bioprocesses. However, the role of m⁶A in heart regeneration remains largely unclear. To study m⁶A function in heart regeneration, we modulated Mettl3 expression in vitro and in vivo. Knockdown of *Mettl3* significantly increased the proliferation of cardiomyocytes and accelerated heart regeneration following heart injury in neonatal and adult mice. However, *Mettl3* overexpression decreased cardiomyocyte proliferation and suppressed heart regeneration in postnatal mice. Conjoint analysis of methylated RNA immunoprecipitation sequencing (MeRIP-seq) and RNA-seq identified *Fgf16* as a downstream target of Mettl3-mediated m⁶A modification during postnatal heart regeneration. RIP-qPCR and luciferase reporter assays revealed that Mettl3 negatively regulates *Fgf16* mRNA expression in an m⁶A-Ythdf2-dependent manner. The silencing of *Fgf16* suppressed the proliferation of cardiomyocytes. However, the overexpression of ΔFgf16, in which the m⁶A consensus sequence was mutated, significantly increased cardiomyocyte proliferation and accelerated heart regeneration in postnatal mice compared with wild-type Fgf16. Our data demonstrate that Mettl3 post-transcriptionally reduces *Fgf16* mRNA levels through an m⁶A-Ythdf2-dependen pathway, thereby controlling cardiomyocyte proliferation and heart regeneration.

## Editor's evaluation

This study explores a novel mechanistic insight into the epitranscriptomic regulation of murine heart regeneration. The authors have demonstrated that Mettl3 post-transcriptionally reduces Fgf16 mRNA levels through an m6 A-Ythdf2-dependent pathway, thereby restricting cardiomyocyte

proliferation and heart regeneration in postnatal mice. Targeting the Mettl3/m6A/Ythdf2/Fgf16 pathway may represent a promising therapeutic strategy to promote the proliferation of cardiomyocytes in mammals.

## Introduction

Heart failure is a leading cause of morbidity and mortality worldwide, due to the inability to regenerate the injured heart after myocardial infarction (*Narula et al., 1996*; *Bergmann et al., 2009*; *Bui et al., 2011*). Although various strategies including cell-based and cell free therapies are being explored to promote heart regeneration in animal models and human patients (*Lin and Pu, 2014*; *Sahara et al., 2015*; *Arrell et al., 2020*; *Biressi et al., 2020*), the efficacy of cardiac therapy and its clinical implications remain uncertain (*Chong et al., 2014*). It has been demonstrated that complete cardiac regeneration occurs in neonatal mouse heart after ventricular resection at postnatal day 1 (p1), but this capacity is lost at p7 (*Porrello et al., 2011*). It has been believed that cardiac regeneration in neonatal mice is primarily mediated by cardiomyocyte proliferation (*Porrello et al., 2011*; *Mahmoud et al., 2013*; *Porrello et al., 2013*). Therefore, it is crucial to explore the related signaling networks and underlying molecular mechanisms that are responsible for postnatal cardiomyocyte proliferation. Previously, many studies have focused on the signaling pathways contributing to the activation of proliferative factors that regulate gene expression in hearts (*Mahmoud et al., 2013*; *Porrello et al., 2013*; *Feng et al., 2019*; *Wu et al., 2021*). However, the potential roles of the posttranscriptional regulation of cardiac mRNAs in cardiac physiology and pathology are largely unknown. Previous studies have reported that mRNA abundance does not correlate with protein expression in failing hearts (*Brundel et al., 2001*; *Su et al., 2015*). Moreover, nuclear and cytosolic mRNA levels are not correlated in cardiomyocytes (*Preissl et al., 2015*). Therefore, these reports suggest that epitranscriptomic regulation plays an important role in healthy and pathological hearts.

Although more than 150 distinct chemical marks have been identified on cellular RNA (*Boccaletto et al., 2018*), the $N^6$-methyladenosine (m6A) methylation catalyzed by the enzyme methyltransferase-like 3 (Mettl3) has been the most prevalent and best-characterized mRNA modification (*Dominissini et al., 2012*; *Meyer et al., 2012*; *Fu et al., 2014*). Recent studies have demonstrated that m6A plays an important role in gene regulation (*Roundtree et al., 2017*), animal development (*Zhao et al., 2017*; *Frye et al., 2018*), and human disease (*Hsu et al., 2017*; *Chen et al., 2018*; *Paris et al., 2019*). A number of studies have demonstrated that effectors of m6A pathways consist of 'writers', 'erasers', and 'readers' that respectively install, remove, and recognize methylation (*Shi et al., 2019*). Growing evidence has revealed that m6A effectors in different biological systems are multifaced and tunable because their expression and cellular localization are cell type- and environmental stimuli-dependent (*Dorn et al., 2019*; *Shi et al., 2019*). Therefore, the mechanisms by which m6A regulate gene expression are complex and context-dependent.

Despite m6A mRNA methylation has been recognized as a posttranscriptional modification in multiple organisms including plants, yeast, flies, and mammals (*Batista et al., 2014*; *Luo et al., 2014*; *Haussmann et al., 2016*; *Wei et al., 2018*), the roles of this posttranscriptional process and functions of m6A mRNA methylation in cardiomyocyte proliferation and animal models of heart regeneration remain unknown. Here, we elucidate the landscape of m6A mRNA methylation during heart regeneration in neonatal mice and show that inhibition of the methylase Mettl3 promotes cardiomyocyte proliferation and heart regeneration. We further demonstrate that posttranscriptional downregulation of Fgf16 by Mettl3-mediated m6A modification via a YT521-B homology (YTH) domain family 2 (Ythdf2)-dependent pathway, inhibits heart regeneration in neonatal mice.

## Results

### Mettl3 expression and the m6A levels in mRNA are increased upon neonatal heart injury

It has been demonstrated that Mettl3-mediated m6A is enhanced in response to hypertrophic stimulation and is necessary for a normal cardiomyocyte hypertrophy in adult mice (*Dorn et al., 2019*), indicating the pivotal functions of Mettl3-mediated m6A in cardiac homeostasis. These further prompt

**eLife digest** Cardiovascular diseases are one of the world's biggest killers. Even for patients who survive a heart attack, recovery can be difficult. This is because – unlike some amphibians and fish – humans lack the ability to produce enough new heart muscle cells to replace damaged tissue after a heart injury. In other words, the human heart cannot repair itself.

Molecules known as messenger RNA (mRNA) carry the 'instructions' from the DNA inside the cell nucleus to its protein-making machinery in the cytoplasm of the cell. These messenger molecules can also be altered by different enzymes that attach or remove chemical groups. These modifications can change the stability of the mRNA, or even 'silence' it altogether by stopping it from interacting with the protein-making machinery, thus halting production of the protein it encodes.

For example, a protein called Mettl3 can attach a methyl group to a specific part of the mRNA, causing a reversible mRNA modification known as $m^6A$. This type of alteration has been shown to play a role in many conditions, including heart disease, but it has been unclear whether $m^6A$ could also be important for the regeneration of heart tissue.

To find out more, Jiang, Liu, Chen et al. studied heart injury in mice of various ages. Newborn mice can regenerate their heart muscle for a short time, but adult mice lack this ability, which makes them a useful model to study heart disease.

Analyses of the proteins and mRNAs in mouse heart cells confirmed that both Mettl3 and $m^6A$-modified mRNAs were present. The amount of each also increased with age. Next, experiments in genetically manipulated mice revealed that removing Mettl3 greatly improved tissue repair after heart injury in both newborn and adult mice. In contrast, mouse hearts that produced abnormally high quantities of Mettl3 were unable to regenerate – even if the mice were young. Moreover, a detailed analysis of gene activity revealed that Mettl3 was suppressing heart regeneration by decreasing the production of a growth-promoting protein called FGF16.

These results reveal a key biological mechanism controlling the heart's ability to repair itself after injury. In the future, Jiang et al. hope that Mettl3 can be harnessed for new, effective therapies to promote heart regeneration in patients suffering from heart disease.

us to ask whether $m^6A$ modification influences heart regeneration in neonatal mice. To determine the expression patterns of $m^6A$ methyltransferases (Mettl3 and Mettl14) and demethylase (Alkbh5 and Fto), myocardium isolated from neonatal hearts at a series of time points was subjected to quantitative real-time PCR (qPCR) assay. Among these four genes, Mettl3 expression significantly increased from postnatal day 3 (p3) to p14 (*Figure 1A*). In agreement with qPCR result, increased expression of Mettl3 in postnatal heart at p14 was also detected in protein levels (*Figure 1B and C*). To determine the cardiac mRNAs modified by $m^6A$ methylation in neonatal mice, $m^6A$ levels were measured by an ELISA-based quantification assay at p3, p7, and p14. Increased levels of $m^6A$ modification were detected at p14 compared with p3 and p7 (*Figure 1D*), suggesting the lower levels of $m^6A$ modification within postnatal 7 days (the regenerative window). These results indicate the potential functional importance of the $m^6A$ modification of cardiac mRNAs for heart regeneration. To further determine the $m^6A$ modification in injured hearts, apical resection was performed at p3, followed by $m^6A$ level examination in sham and resected hearts at 5 days post-resection (dpr) (*Figure 1E*). As shown in *Figure 1F*, apex resection injury increased the level of $m^6A$ modification by 65% at 5 dpr compared with sham hearts. Moreover, the expression of *Mettl3* was approximately increased up to 2-fold at 5 dpr compared with sham hearts (*Figure 1G*). Western blotting revealed that the expression of Mettl3 rather than Mettl14 was greatly increased at 5 dpr compared with sham hearts (*Figure 1H–J*). In consistent with apical areas, our data showed that Mettl3 expression and $m^6A$ levels were also increased in remote areas at 5 dpr compared with control groups (*Figure 1—figure supplement 1*). To further determine the $m^6A$ levels in different cardiac cell types, primary cardiomyocytes (CMs) and non-cardiomyocytes (nCMs) were isolated from neonatal heart at p1 according to the method previously reported (*Wu et al., 2021*), followed by RNA extraction and $m^6A$ level determination using colorimetric ELISA assay. Our data showed that the $m^6A$ level in CMs is higher than that in nCMs, indicating the functional importance of $m^6A$ modification in CMs (*Figure 1—figure supplement 2A*). To determine the response of Mettl3 expression in CMs and nCMs to injury, primary CMs and nCMs

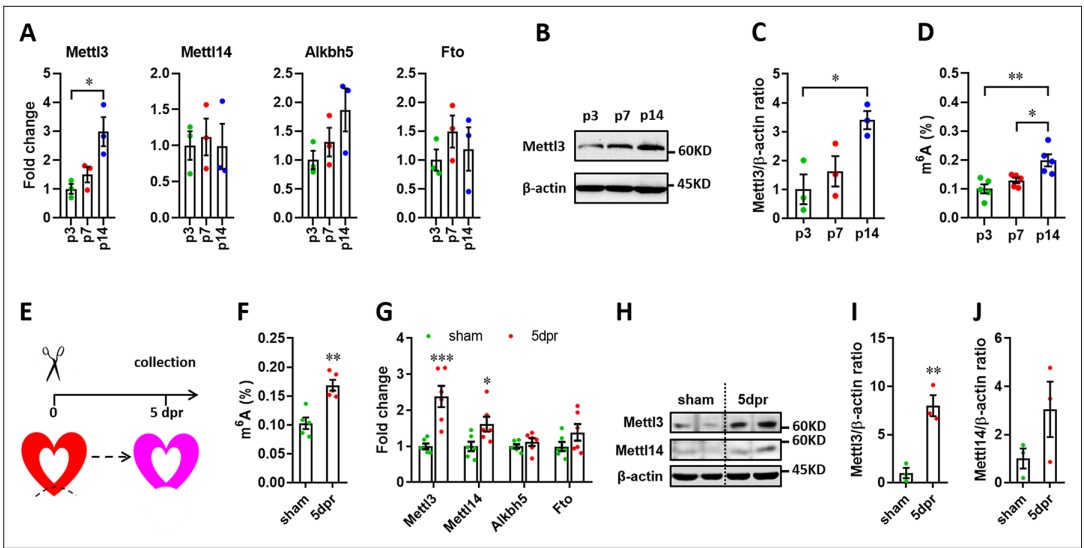

**Figure 1.** Expression patterns of m⁶A methylases and demethylases during heart regeneration in neonatal mice. (**A**) Quantification of m⁶A methylases (Mettl3 and Mettl14) and demethylases (Alkbh5 and Fto) expression in neonatal hearts at the indicated time points after birth (n=3 hearts). (**B and C**) Representative images (**B**) and quantification (**C**) of protein expression of Mettl3 in neonatal hearts (n=3 hearts). (**D**) Cardiac mRNA m⁶A levels in neonatal heart were measured by ELISA-based quantification assay (n=5 hearts per group). (**E**) Schematic of heart injury and sample collection in neonatal mice. (**F**) mRNA m⁶A levels in the injured hearts at 5 dpr were measured (n=5 hearts). (**G**) qPCR validation of m⁶A methylases and demethylases in neonatal hearts at 5 dpr (n=6 hearts). (**H–J**) Protein levels of Mettl3 and Mettl14 were measured by western blotting at 5 dpr. Representative images (**F**) and quantification (**G and H**) of protein expression of Mettl3 and Mettl14 are shown (n=3 hearts). All data are presented as the mean ± SEM. *p<0.05, **p<0.01, ***p<0.001 compared with p3 (**A and C**) or sham (**F, G, and I**). p values were determined by 1-way (A, C, and D) or 2-way (**G**) ANOVA with Dunnett's multiple-comparison test, or by 2-tailed Student's t-test (F, I, and J).

The online version of this article includes the following source data and figure supplement(s) for figure 1:

**Source data 1.** Western blot for *Figure 1B* showing Mettl3 expression.

**Source data 2.** Western blot for *Figure 1B* showing β-actin expression.

**Source data 3.** Western blot for *Figure 1H* showing Mettl3 expression.

**Source data 4.** Western blot for *Figure 1H* showing Mettl14 expression.

**Source data 5.** Western blot for *Figure 1H* showing β-actin expression.

**Figure supplement 1.** Mettl3 expression and m⁶A modification levels in the remote areas of injured neonatal heart at 5 dpr.

**Figure supplement 1—source data 1.** Western blot for *Figure 1—figure supplement 1A* showing Mettl3 expression.

**Figure supplement 1—source data 2.** Western blot for *Figure 1—figure supplement 1A* showing β-actin expression.

**Figure supplement 2.** M⁶A modification and Mettl3 expression levels in CMs and nCMs.

were isolated from sham and injured neonatal hearts at 5 dpr. Our data showed that Mettl3 expression was significantly increased in CMs rather than nCMs upon injury (*Figure 1—figure supplement 2B*). Taken together, these findings suggest that Mettl3-mediated m⁶A modification might be important for regulating heart regeneration.

## In vitro effects of Mettl3 on the proliferation of cardiomyocytes

Previous studies have demonstrated that cardiomyocyte proliferation is vital to heart regeneration (*Porrello et al., 2011*; *Mahmoud et al., 2013*). This idea promoted us to ask whether Mettl3-mediated m⁶A also influences cardiomyocyte proliferation. In the rat cardiomyocyte cell line (H9c2), we silenced the expression of *Mettl3* gene using siRNA (si*Mettl3*) (*Figure 2—figure supplement 1A*) and performed nuclear incorporation assay with 5-ethynyl-2'-deoxyuridine (EdU). The percentage of

EdU[+] cells was significantly increased by *Mettl3* knockdown (*Figure 2—figure supplement 1B, C*), suggesting elevated levels of proliferation in *Mettl3*-silencing cells. Indeed, cell number was also greatly promoted by siMettl3 (*Figure 2—figure supplement 1D*). To further explore the potential effects of Mettl3 overexpression on H9c2 cell proliferation, a stable cell line was established using lentiviral-mediated overexpression of Mettl3 (*Figure 2—figure supplement 1E-H*). As expected, increased m[6]A levels were observed in the Mettl3-overexpressing cells compared with controls (*Figure 2—figure supplement 1I, J*). In contrast with the knockdown cells, both cell proliferation and count were remarkably suppressed by Mettl3 overexpression in H9c2 cells (*Figure 2—figure*

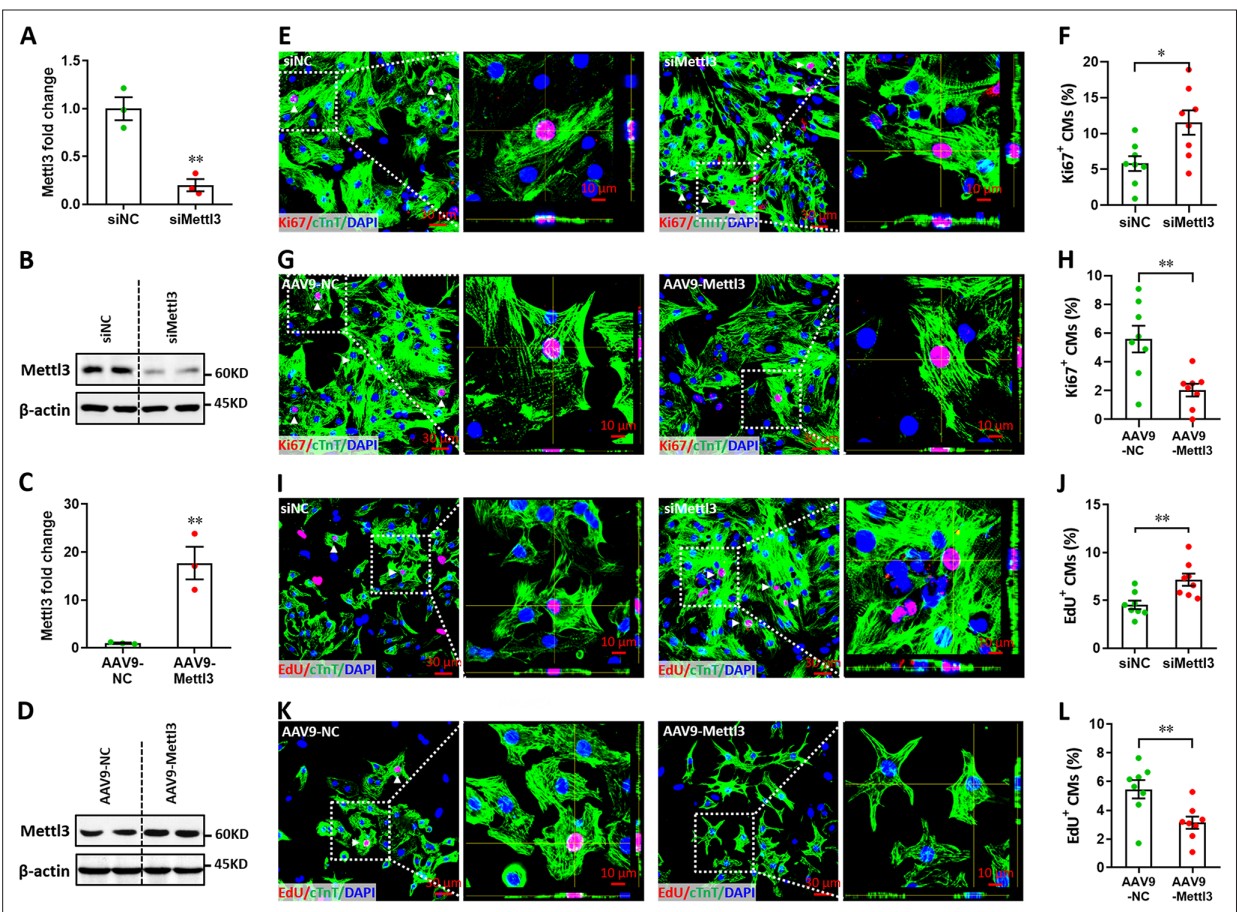

**Figure 2.** Mettl3 suppresses the proliferation of primary cardiomyocytes in vitro. (**A and B**) Primary cardiomyocytes were transfected with si*Mettl3* and si*NC* for 48 hr, followed by qPCR (A, n=3) and western blotting (**B**) validation. (**C and D**) Primary cardiomyocytes were transfected with AAV9-NC and AAV9-Mettl3 for 72 hr, followed by qPCR (C, n=3) and western blotting (**D**) validation. (**E–H**) Mettl3 silencing and overexpressing cardiomyocytes were subjected to Ki67 (red) and cTnT (green) double staining. Representative images (**E, G**) and quantification (**F, H**) of Ki67[+] cardiomyocytes are shown (n=8). Magnified Z-stack confocal images of Ki67[+] cardiomyocytes are shown in right panel (**E, G**). (**I–L**) Mettl3 silencing and overexpressing cardiomyocytes were subjected to EdU (red) and cTnT (green) double staining. Representative images (**I, K**) and quantification (**J, L**) of EdU[+] cardiomyocytes are shown (n=8). Magnified Z-stack confocal images of EdU[+] cardiomyocytes are shown in right panel (**I, K**). All data are presented as the mean ± SEM of three separate experiments, *p<0.05, **p<0.01 versus control. p values were determined by 2-tailed Student's t-test.

The online version of this article includes the following source data and figure supplement(s) for figure 2:

**Source data 1.** Western blot for *Figure 2B* showing Mettl3 expression.

**Source data 2.** Western blot for *Figure 2B* showing β-actin expression.

**Source data 3.** Western blot for *Figure 2D* showing Mettl3 expression.

**Source data 4.** Western blot for *Figure 2D* showing β-actin expression.

**Figure supplement 1.** Effects of Mettl3 on the proliferation of H9c2 cells.

**Figure supplement 1—source data 1.** Western blot for *Figure 2—figure supplement 1F* showing Mettl3 expression.

**Figure supplement 1—source data 2.** Western blot for *Figure 2—figure supplement 1F* showing β-actin expression.

*supplement 1K-M*). These results reveal that cardiomyocyte proliferation might be negatively controlled by Mettl3-mediated m⁶A modification.

To further confirm this idea, primary cardiomyocytes isolated from neonatal mice were subjected to proliferation assay. Ki67 and cTnT double staining revealed that *Mettl3* knockdown (*Figure 2A and B*) leads to an increase in the percentage of Ki67⁺ cTnT⁺ cells, indicating an elevated proliferation level of primary cardiomyocytes (*Figure 2E and F*). In contrast, *Mettl3* overexpression (*Figure 2C and D*) suppressed the proliferation of primary cardiomyocytes as demonstrated by Ki67 and cTnT double staining (*Figure 2G and H*). In agreement with these results, EdU incorporation assay revealed that the percentage of EdU⁺ cTnT⁺ cells was greatly increased by *Mettl3* knockdown (*Figure 2I and J*), indicating an elevated proliferation level of primary cardiomyocytes. However, the proliferation of primary cardiomyocytes was approximately suppressed by 48% by *Mettl3* overexpression as demonstrated by nuclear EdU incorporation (*Figure 2K and L*). Taken together, these findings from primary cardiomyocytes and cell line suggest that Mettl3-mediated m⁶A modification can negatively regulate the proliferation of cardiomyocytes.

## In vivo knockdown of Mettl3 promotes cardiomyocyte proliferation in injured neonatal hearts

To knock down Mettl3 in myocardium of neonatal mice, adeno-associated virus 9 (AAV9)-mediated delivery of shRNA was used. First, the time course efficiency of AAV9-sh*RNA* system was determined in neonatal heart using the harboring EGFP reporter gene. Our results revealed that AAV9-sh*RNA* system induces modest expression of reporter gene in neonatal heart at 2 days post-injection (dpi), followed by substantial expression from 6 to 27 dpi (*Figure 3—figure supplement 1*). To investigate the in vivo roles of Mettl3 in cardiomyocyte proliferation, AAV9-sh*Mettl3* viruses were injected into neonatal hearts at p1, followed by apex resection at p3 (equal to 0 dpr) and a nuclear EdU incorporation assay at 5 dpr (*Figure 3A*). Successful knockdown of Mettl3 in heart tissue at 0 dpr was confirmed by the qPCR validation of Mettl3 mRNA expression (*Figure 3B*). To further confirm Mettl3 knockdown, primary cardiomyocytes isolated from ventricles at 0 dpr were subjected to western blotting and ELISA-based m⁶A quantification assay. The decreased expression of Mettl3 protein (*Figure 3C and D*) and reduced m⁶A modification (*Figure 3E*) was further demonstrated in primary cardiomyocytes. To determine the size of the cardiomyocytes in neonatal heart at 5 dpr, wheat germ agglutinin (WGA) staining was performed. Smaller cardiomyocyte size with no change in heart to body weight ratio was observed in Mettl3-silencing hearts compared with controls (*Figure 3F–H*), implying increased numbers of cardiomyocytes in Mettl3-deficient hearts. These results suggest that in vivo deficiency of Mettl3 promotes heart regeneration, implying elevated proliferation potential of cardiomyocytes.

To further confirm this conjecture, in vivo proliferation of cardiomyocytes was further examined at 5 dpr. It has been demonstrated that cardiomyocytes can go through cell cycle phases but stop and not complete it (*Lázár et al., 2017*). To prove complete cardiomyocyte proliferation, late cell cycle markers including phospho-Histone H3 (pH3) and aurora kinase B (AurkB) were further used in this study. Consistently, the percentage of pH3⁺ cTnT⁺ cells was significantly elevated by Mettl3 silencing both in apical (*Figure 3I and J*) and remote (*Figure 3—figure supplement 2A, B*) zones, indicating that Mettl3 knockdown really promotes the complete proliferation of cardiomyocytes in vivo. Importantly, increased percentage of AurkB⁺ cardiomyocytes was also detected in cardiac apex in the Mettl3-deficient neonatal mice (*Figure 3K–M*), implying the increased cytokinesis of cardiomyocytes. In agreement with above observation, in vivo staining of Ki67 further revealed that cardiomyocyte proliferation in the injured neonatal hearts was greatly increased by Mettl3 silencing in the apical (*Figure 3N and O*) and remote (*Figure 3—figure supplement 2C, D*) zones compared with controls. Moreover, nuclear EdU incorporation revealed that in vivo silencing of *Mettl3* increased the percentage of EdU⁺ cTnT⁺ cells both in apical (*Figure 3P and Q*) and remote (*Figure 3—figure supplement 2E, F*) cardiac tissues. These data suggest that Mettl3-mediated m⁶A modification might negatively controls cardiomyocyte proliferation during neonatal heart regeneration. Taken together, these in vitro and in vivo findings strongly demonstrated that Mettl3-mediated m⁶A modification can negatively regulate cardiomyocyte proliferation during heart regeneration in neonatal mice.

To further explore whether Mettl3 knockdown influences cardiomyocyte proliferation in the homeostatic neonatal hearts without injury, AAV9-sh*Mettl3* and control viruses were injected at p1, followed by sample collection and histological analysis at p8 (*Figure 3—figure supplement 3A*). This virus

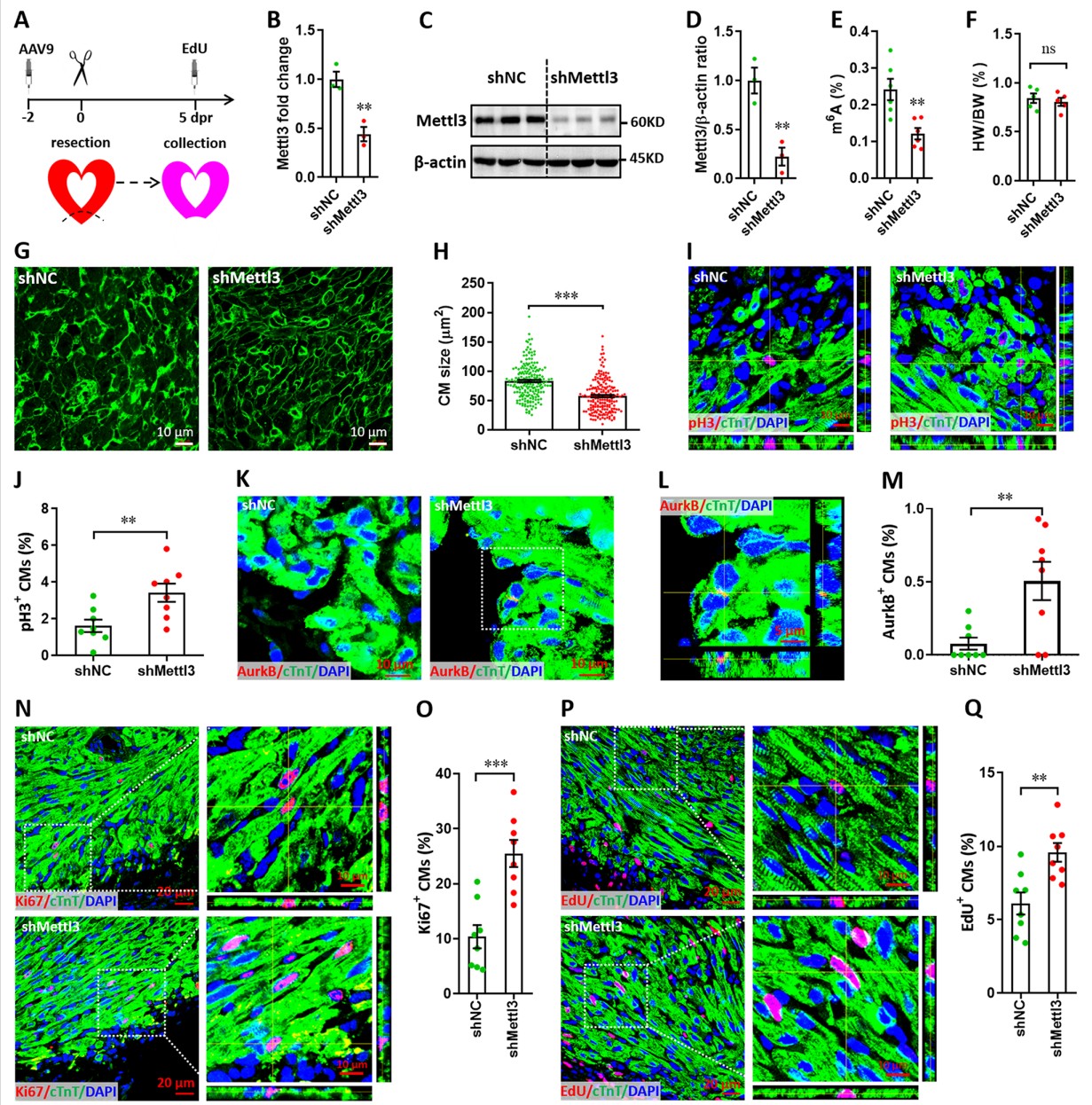

**Figure 3.** Mettl3 knockdown promotes cardiomyocyte proliferation in neonatal hearts at 5 dpr. (**A**) Schematic of AAV9-sh*Mettl3* virus injection designed to knock down *Mettl3* in neonatal hearts. (**B**) qPCR validation of *Mettl3* in the AAV9-injected hearts at 0 dpr (n=3 hearts). (**C–E**) Primary cardiomyocytes were isolated from neonatal hearts with AAV9 virus injection at 0 dpr, followed by western blotting and m6A quantification. Representative images (**C**) and quantification (D, n=3) of Mettl3 protein expression, as well as the quantification of m6A modification levels (E, n=6) are shown. (**F**) Quantification of heart weight (HW) to body weight (BW) ratio (n=5 hearts). (**G and H**) Representative WGA staining images (**G**) and quantification (**H**) of the size of cardiomyocytes located in border zone at 5 dpr (n=~200 cells from 5 hearts per group). (**I and J**) Representative Z-stack confocal images (**I**) and quantification (**J**) of pH3+ cardiomyocytes in apical ventricle at 5 dpr (n=8 hearts). (**K–M**) Representative images (**K and L**) and quantification (**M**) of AurkB+ cardiomyocytes in apical zone at 5 dpr (n=8 hearts). L, magnified Z-stack confocal image in K (right panel). (**N and O**) Representative images (**N**) and quantification (**O**) of Ki67+ cardiomyocytes in apical ventricle at 5 dpr (n=8 hearts). Right panel (**N**), magnified Z-stack confocal images. (**P and Q**) Representative images (**P**) and quantification (**Q**) of EdU+ cardiomyocytes in apical ventricle at 5 dpr (n=8 hearts). Right panel (**P**), magnified Z-stack confocal images. All data are presented as the mean ± SEM, *p<0.05, **p<0.01, ***p<0.001 versus control. p values were determined by 2-tailed Student's t-test.

The online version of this article includes the following source data and figure supplement(s) for figure 3:

**Source data 1.** Western blot for **Figure 3C** showing Mettl3 expression.

**Source data 2.** Western blot for **Figure 3C** showing β-actin expression.

*Figure 3 continued on next page*

*Figure 3 continued*

**Figure supplement 1.** Time course of AAV9 system infection in neonatal mice heart.

**Figure supplement 2.** Mettl3 knockdown promotes cardiomyocyte proliferation in the remote area in injured heart at 5 dpr.

**Figure supplement 3.** Effects of Mettl3 deficiency on cardiomyocyte proliferation in neonatal mice without injury at p8.

injection strategy and expression time are equal to the time point of 5 dpr in the apex resection injury model as mentioned in *Figure 3A*. Our data showed that Mettl3 knockdown has no significant effects on heart weight/body weight (HW/BW) ratio compared with controls (*Figure 3—figure supplement 3B*). WGA staining revealed that cardiomyocyte size is comparable between Mettl3-silencing and control hearts (*Figure 3—figure supplement 3C, D*). Moreover, no significant differences in cardiomyocyte proliferation were observed in Mettl3-silencing hearts compared with controls, which were determined by EdU, Ki67, and pH3 staining (*Figure 3—figure supplement 3E, J*). These findings suggest that Mettl3 knockdown has no significant effects on cardiomyocyte proliferation in the uninjured neonatal hearts.

## Mettl3 knockdown accelerates heart regeneration in neonatal mice

It is well known that neonatal mice possess the capacity of heart regeneration upon injury within p7. However, neonatal mouse could not completely finish the heart regeneration until 21 days post-injury even induced at p1 (*Porrello et al., 2011*; *Mahmoud et al., 2013*; *Porrello et al., 2013*). To further investigate whether Mettl3 deficiency indeed promotes heart regeneration, AAV9-sh*Mettl3* viruses were injected into neonatal mice at p1, followed by apex resection at p3 and sample collection at 14 dpr, a time point prior to the completion of heart regeneration (*Figure 4A*). Histological analysis revealed that scar size in the cardiac apex at 14 dpr was decreased by 73% in Mettl3-deficient hearts compared with controls (*Figure 4B and C*), implying the accelerated heart regeneration. In agreement with these results, smaller scar sizes in Mettl3-deficient hearts were also confirmed by whole heart images (*Figure 4D–F*). In addition, cardiomyocyte size was much smaller in Mettl3-deficient hearts compared with controls (*Figure 4G and H*). These results suggest that Mettl3 deficiency leads to fast regeneration of neonatal heart as early as 14 dpr. As expected, increased left ventricular ejection fraction (LVEF) and fractional shortening (LVFS) were detected in Mettl3-deficient mice (*Figure 4I–K*), indicating that left ventricular systolic function was promoted by Mettl3 knockdown. To determine proliferating cardiomyocytes during the whole period of heart regeneration within 14 dpr, EdU was injected in a pulse chase manner (*Figure 4A*). Our data revealed that the percentage of EdU$^+$ cTnT$^+$ cells both in apical and remote zones were significantly increased in Mettl3-deficient hearts compared with controls (*Figure 4L–O*), implying that there are more proliferating cardiomyocytes during the whole period of 14 dpr. Taken together, these data suggest that Mettl3 knockdown promotes cardiomyocyte proliferation in the early stage of heart regeneration, thereby accelerating heart regeneration within two weeks post-resection in neonatal mice. In contract, Mettl3 knockdown did not lead to significant changes in cardiomyocyte size, proliferation, as well as cardiac function in the homeostatic neonatal mice without injury (*Figure 4—figure supplement 1*).

 To further explore whether Mettl3 knockdown extends the regenerative window, AAV9-shMettl3 viruses were injected at p1, followed by apex resection at p7, when the regenerative potential has mostly ceased (*Figure 4—figure supplement 2A*). Mettl3 knockdown at p7 was confirmed by western blotting (*Figure 4—figure supplement 2B, C*). Moreover, Mettl3 knockdown-induced decrease in m$^6$A levels at p7 was further demonstrated by ELISA-based m$^6$A quantification assay (*Figure 4—figure supplement 2D*). To determine the proliferation of cardiomyocytes at 3 dpr (the early stage of heart injury), immunofluorescence staining was performed using antibodies against Ki67 or pH3 together with cTnT antibodies. Ki67 staining revealed that cardiomyocyte proliferation in the apical zones of injured neonatal hearts was significantly increased by Mettl3 knockdown (*Figure 4—figure supplement 2E-G*). In addition, Mettl3-silencing increased the percentage of pH3$^+$ cTnT$^+$ cells in apical myocardium (*Figure 4—figure supplement 2H-J*). These results are agreement with the histological analysis at 21 dpr which revealed that scar size in the cardiac apex was decreased in Mettl3-deficient hearts compared with controls (*Figure 4—figure supplement 2K*), implying the improved heart regeneration. Moreover, improving cardiac function was detected in Mettl3-deficient heart at 21 dpr as demonstrated by the increased LVEF and LVFS values (*Figure 4—figure supplement 2L-N*).

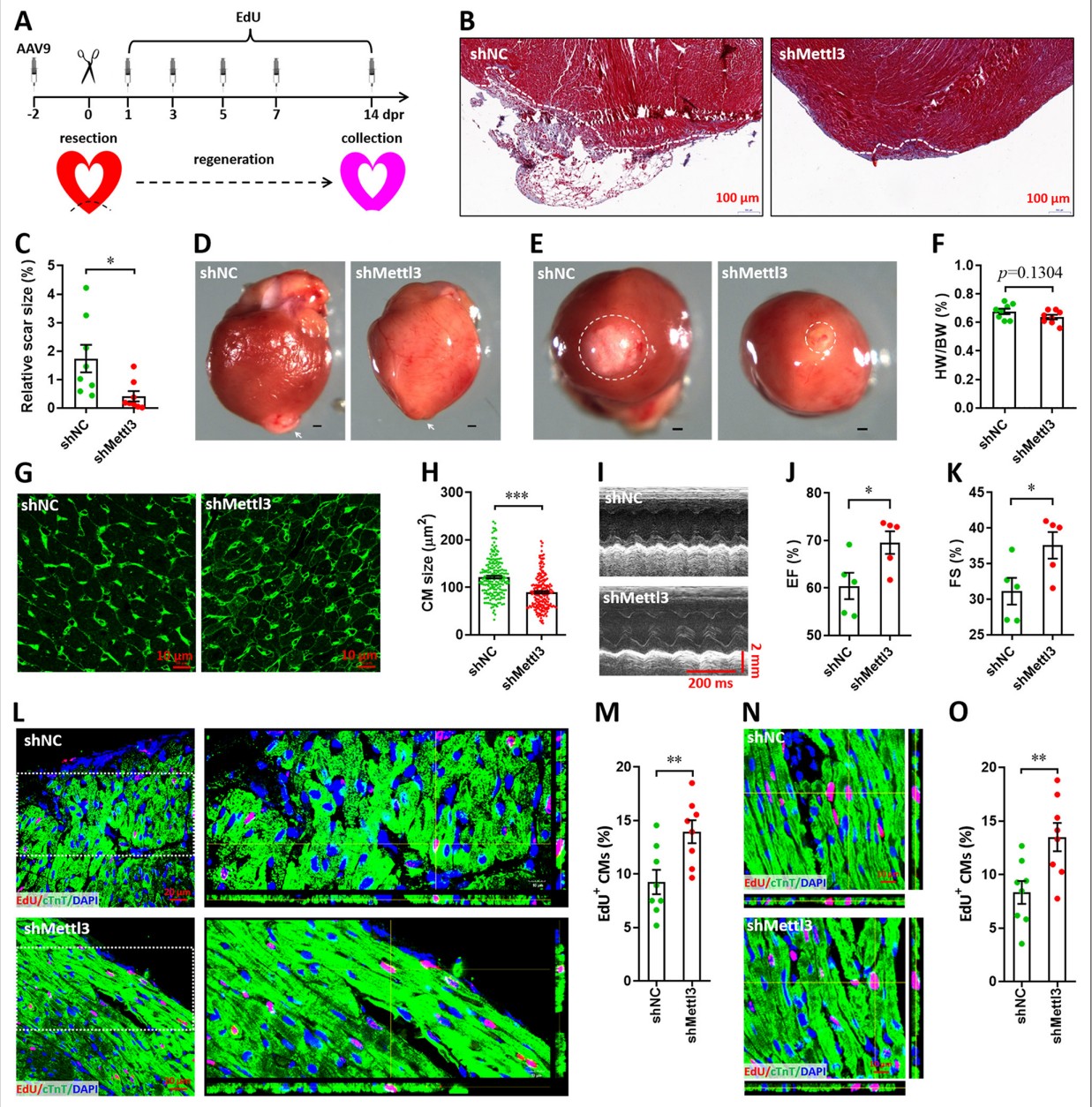

**Figure 4.** Knockdown of Mettl3 accelerates heart regeneration in neonatal mice at 14 dpr. (**A**) Schematic of AAV9 virus injection, apex resection, and EdU-pulse injection, followed by sample collection at 14 dpr. (**B and C**) Representative masson's trichrome staining images of cardiac apex (**B**) and quantification of scar size (**C**) in control and Mettl3-deficient hearts (n=8 hearts). (**D and E**) Representative whole (**D**) and apical (**E**) images of neonatal hearts at 14 dpr. Arrowhead (**D**) and circle (**E**) denote scars in cardiac apex. (**F**) Quantification of heart weight (HW) to body weight (BW) ratio (n=8 hearts). (**G**) Representative WGA staining images of myocardium located in border zone in control and Mettl3-deficient hearts. (**H**) Quantification of cardiomyocyte size in control and Mettl3 knockdown hearts are shown (n=~200 cells from 5 hearts). (**I–K**) Representative images of M-model echocardiography (**I**) and quantification of LVEF (**J**) and LVFS (**K**) are shown (n=5 hearts). (**L**) Representative images of EdU+ cardiomyocytes (double positive for EdU and cTnT) in the injured cardiac apex. Right panel, representative Z-stack confocal images. (**M**) Quantification of EdU+ cardiomyocytes in apical zone of control and Mettl3-deficient hearts (n=8 hearts). (**N and O**) Representative Z-stack confocal images (**N**) and quantification (**O**) of EdU+ cardiomyocytes in the uninjured ventricle (remote zone) are shown (n=8 hearts). All data are presented as the mean ± SEM, *p<0.05, **p<0.01, ***p<0.001 versus control. p values were determined by 2-tailed Student's t-test.

The online version of this article includes the following source data and figure supplement(s) for figure 4:

**Figure supplement 1.** Effects of Mettl3 deficiency on heart regeneration in postnatal mice without injury at p17.

**Figure supplement 2.** Mettl3 knockdown promotes heart regeneration upon apex resection at p7.

**Figure supplement 2—source data 1.** Western blot for *Figure 4—figure supplement 2B* showing Mettl3 expression.

*Figure 4 continued on next page*

*Figure 4 continued*

**Figure supplement 2—source data 2.** Western blot for *Figure 4—figure supplement 2B* showing β-actin expression.

Taken together, these findings further suggest that Mettl3 deficiency extends regenerative window in neonatal mice.

## Mettl3 overexpression suppresses heart regeneration in postnatal mice

We subsequently investigated the potential roles of Mettl3 overexpression in heart regeneration. To overexpress Mettl3 in cardiomyocytes, AAV9-Mettl3 viruses were injected into neonatal mice at p1, followed by apex resection at p3 (*Figure 5A*). Overexpression of Mettl3 protein and increase in m⁶A modification at p3 was confirmed by western blotting (*Figure 5B and C*) and ELISA-based m⁶A quantification assay (*Figure 5D*), respectively. To determine the potential inhibition of heart regeneration induced by Mettl3 overexpression, the resected neonatal mice were raised until 28 dpr (*Figure 5A*), when beyond the time to complete regeneration (21 dpr) for neonatal mice (*Porrello et al., 2011*). Persistent overexpression of Mettl3 at 28 dpr was further evidenced by qPCR and immunofluorescent staining (*Figure 5—figure supplement 1A, B*), this was further supported by the substantial expression of EGFP reporter gene (*Figure 5—figure supplement 1C*). Trichrome staining results revealed an increase in fibrotic scar size in Mettl3-overexpressing hearts compared with controls (*Figure 5E–G*). The larger scar size induced by Mettl3 overexpression was further confirmed by whole heart images in parallel with the increased HW/BW ratio (*Figure 5H–J*). In agreement with these results, cardiomyocyte size was increased in response to Mettl3 overexpression at 28 dpr (*Figure 5K and L*). To evaluate the effects of Mettl3 on cardiac function, mice were subjected to echocardiography. Our data showed that Mettl3 overexpression leads to decreases in LVEF and LVFS levels (*Figure 5M–O*), indicating that left ventricular systolic function is attenuated by Mettl3 overexpression.

Using the immunofluorescence staining with antibodies against Ki67 and pH3, no positive cardiomyocytes were detected in both groups at 28 dpr (*Figure 5—figure supplement 2*). However, pulse-chase injection of EdU (*Figure 5P*) revealed that cardiomyocyte proliferation in the apical (*Figure 5Q and S*) and remote (*Figure 5R and T*) zones was remarkably suppressed by Mettl3 overexpression during the whole period of 28 days post-resection. Therefore, these results suggest that Mettl3 overexpression might suppress heart regeneration in neonatal mice by inhibiting cardiomyocyte proliferation in the early stage of hear injury and repair. However, Mettl3 overexpression did not change the HW/BW ratio and cardiomyocyte size in the postnatal mice without injury (*Figure 5—figure supplement 3A-D*). EdU-pulse injection experiments showed that cardiomyocyte proliferation was not influenced by Mettl3 overexpression in the uninjured hearts (*Figure 5—figure supplement 3E, F*). In line with these results, there were no significant changes in cardiac function in the uninjured hearts upon Mettl3 overexpression (*Figure 5—figure supplement 3G, H*).

## Mettl3 knockdown promotes heart regeneration at non-regenerative stages

Subsequently, myocardium infarction (MI) injury model was used to further determine the effects of Mettl3 on heart regeneration in postnatal and adult mice. To determine the response of Mettl3 to heart injury at non-regenerative stages, western blotting was performed using heart tissues from postnatal (p7) and adult (3-month-old) mice at 5 days post-MI (dpM). In contrast with neonatal mice model, heart injury did not significantly increase Mettl3 expression in postnatal and adult mice (*Figure 6—figure supplement 1*). We thus further examined the potential influence of Mettl3 deficiency on heart injury at non-regenerative stages. To silence Mettl3 expression in heart, AAV9-sh*Mettl3* and control viruses were injected into neonatal mice at p1, followed by MI surgery at p7. Samples were then collected and analyzed at 5 and 21 dpM, respectively (*Figure 6A*). WGA staining results revealed that Mettl3 deficiency significantly reduced the size of cardiomyocytes in the infarcted neonatal hearts at 5 dpM (*Figure 6B and C*). Ki67 and cTnT double staining revealed that Mettl3 knockdown promotes the proliferation of cardiomyocytes in the border zone of infarcted neonatal hearts as evidenced by the increased percentage of Ki67⁺ cTnT⁺ cells (*Figure 6D and E*). Moreover, increased percentage of pH3⁺ cTnT⁺ cells was detected in the border zone in the Mettl3-deficient hearts at 5 dpM (*Figure 6F and G*). We further analyzed the cardiac histology at 21 dpM and found that both HW/BW ratio and

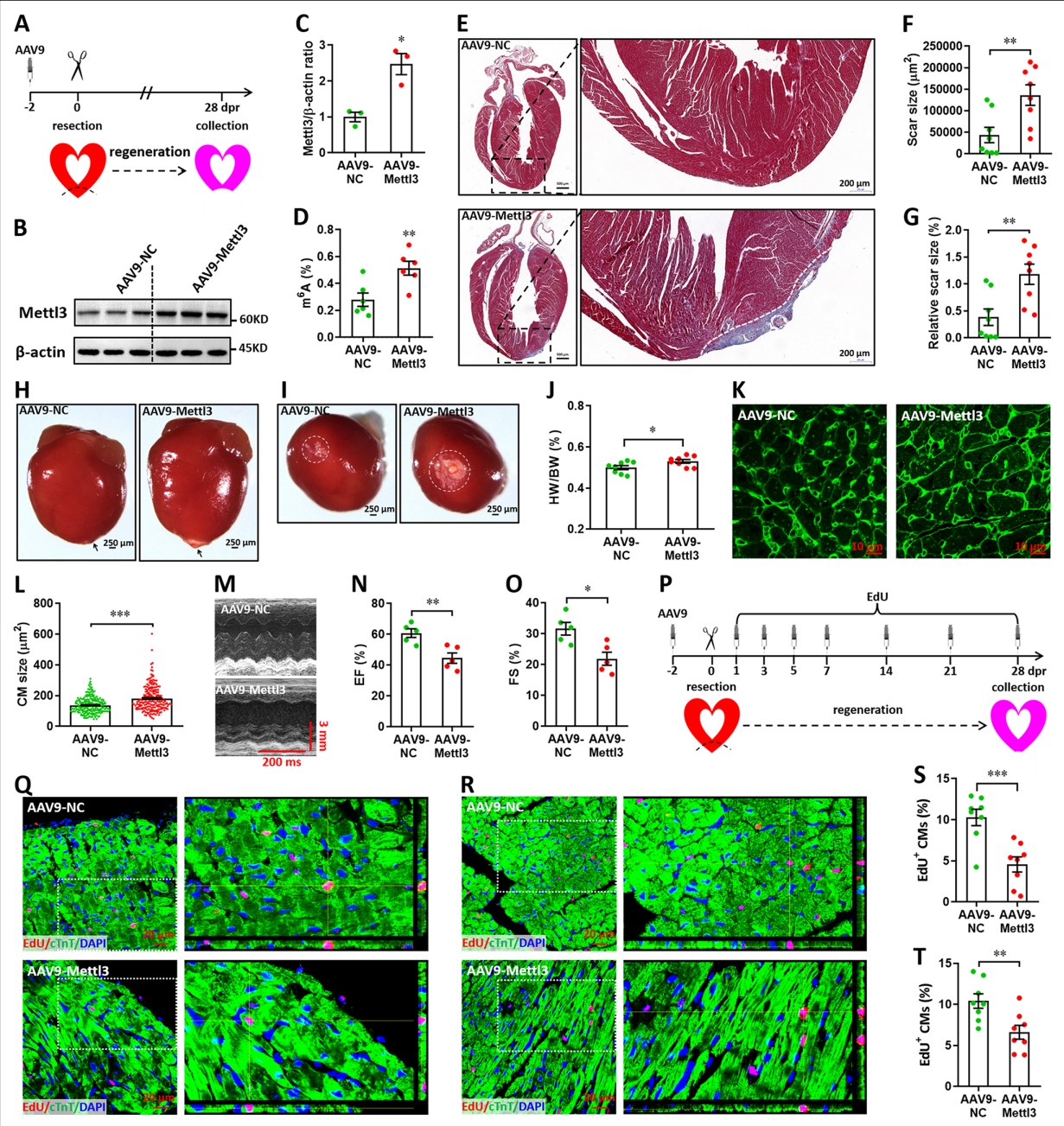

**Figure 5.** Overexpression of Mettl3 suppresses heart regeneration in neonatal mice upon injury. (**A**) Schematic of AAV9-Mettl3 virus injection, apex resection, and sample collection in neonatal mice. (**B and C**) Representative images (**B**) and quantification (**C**) of western blotting for Mettl3 expression in neonatal heart at 0 dpr (n=3 hearts). (**D**) $m^6A$ levels in the heart at 0 dpr were measured by ELISA-based quantification assay (n=6 hearts). (**E**) Representative masson's trichrome staining images of heart in control and Mettl3-overexpressing mice at 28 dpr. (**F and G**) Direct (**F**) and relative (**G**) quantification of scar size in control and Mettl3-overexpressing hearts (n=8 hearts). Relative scar size is presented as percentages of whole ventricle size. (**H and I**) Representative whole (**H**) and apical (**I**) images of neonatal hearts at 28 dpr. Arrowhead (**H**) and circle (**I**) denote scars in cardiac apex. (**J**) Quantification of heart weight (HW) to body weight (BW) ratio (n=8 hearts). (**K and L**) Representative WGA staining images (**K**) and quantification of cardiomyocyte size (**L**) in border zone of control and Mettl3-overexpressing hearts at 28 dpr (n=~200 cells from 5 hearts). (**M–O**) Representative images of M-model echocardiography (**M**) and quantification of LVEF (**N**) and LVFS (**O**) are shown at 28 dpr (n=5 hearts). (**P**) Schematic of EdU pulse injection and sample collection at 28 dpr. (**Q and R**) Representative images of EdU+ cardiomyocytes in apical (**Q**) and remote (**R**) zone at 28 dpr. Right panel, magnified Z-stack confocal images. (**S and T**) Quantification of EdU+ cardiomyocytes in apical (**S**) and remote (**T**) zone in control and Mettl3-overexpressing hearts (n=8 hearts). All data are presented as the mean ± SEM, *p<0.05, **p<0.01, ***p<0.001 versus control. p values were determined by 2-tailed Student's t-test.

The online version of this article includes the following source data and figure supplement(s) for figure 5:

*Figure 5 continued on next page*

*Figure 5 continued*

**Source data 1.** Western blot for *Figure 5B* showing Mettl3 expression.

**Source data 2.** Western blot for *Figure 5B* showing β-actin expression.

**Figure supplement 1.** Overexpression of Mettl3 in postnatal hearts with AAV9 injection at p27.

**Figure supplement 2.** No proliferating cardiomyocytes was detected in heart at 27 dpr.

**Figure supplement 3.** Effects of Mettl3 overexpression on heart regeneration in postnatal mice without injury at p31.

scar size were lower in the Mettl3-deficient heart compared with controls (*Figure 6H–J*). In agreement with these findings, smaller size of cardiomyocytes was also detected in the Mettl3-deficient hearts compared with controls (*Figure 6K and L*). These results suggest that Mettl3 deficiency might attenuate the cardiac remodeling during myocardium infarction in postnatal mice. Echocardiography analysis at 21 dpM showed that Mettl3 deficiency significantly increases LVEF and LVFS levels compared with control hearts (*Figure 6M–O*).

In addition, we further confirmed our conclusion in adult mice model. To knock down the expression of Mettl3 in the heart of adult mice, 8-week-old mice were injected with AAV9-sh*Mettl3* and control viruses, respectively. Mice were then subjected to MI surgery 4 weeks after virus injection, followed by histological analysis and cardiac function analysis at 14 and 28 dpM, respectively (*Figure 7A*). The persistent expression of AAV9-sh*Mettl3* in adult heart after virus injection for 4 weeks (equal to 0 dpM) was validated by the immunofluorescent staining for EGFP (*Figure 7B*). At 14 dpM, HW/BW ratio and scar size were decreased by Mettl3 deficiency compared with controls (*Figure 7C–E*). In line with these results, the decreased HW/BW ratio and scar size were also observed in the Mettl3-deficient hearts compared with controls at 28 dpM (*Figure 7F–H*). These results suggest that Mettl3 deficiency might attenuate the cardiac remodeling during myocardium infarction. To further evaluate the effects of Mettl3 deficiency on cardiac function, mice were subjected to echocardiography analysis at 0 and 28 dpM, respectively. Our data showed that Mettl3 deficiency leads to a little increase in LVEF and LVFS levels at 0 dpM (prior to MI). However, increased levels of LVEF and LVFS were detected in Mettl3-deficient hearts at 28 dpM compared with control hearts, respectively (*Figure 7I–K*). These results indicate that Mettl3 deficiency can significantly attenuate the cardiac dysfunction induced by MI in adult mice, but this benefit is not sensitive in homeostatic hearts.

To further determine whether Mettl3 deficiency influences the proliferation of cardiomyocytes in adult mice during MI injury, immunofluorescent staining was performed in cardiac tissues at 14 dpM. Ki67 staining results showed that Mettl3 deficiency greatly increased the proliferation of cardiomyocytes in the border zone of infarction, as indicated by the increased percentage of Ki67$^+$ cTnT$^+$ cells (*Figure 7L and M*). In consistent with border zone, increased percentage of Ki67$^+$ cTnT$^+$ cells was also detected in the Mettl3-deficient hearts in the remote zone in comparison with control hearts (*Figure 7N and O*). These above findings imply that Mettl3 deficiency promotes cardiac repair and regeneration in postnatal mice at the non-regenerative stages, at least in part, through promoting cardiomyocyte proliferation.

## Variations in m$^6$A-regulated heart genes upon injury

To investigate target genes involved in both Mettl3-mediated m$^6$A modification and heart regeneration, ventricles from sham and 5 dpr hearts were subjected to methylated RNA immunoprecipitation sequencing (MeRIP-seq) and RNA-seq analysis. MeRIP-seq revealed that the GGACU motif is highly enriched in m$^6$A sites in both sham and apex-resected hearts (*Figure 8—figure supplement 1A*). Moreover, m$^6$A peaks are particularly abundant in the vicinity of stop codons (*Figure 8—figure supplement 1B*). However, the m$^6$A peak distribution decreased in 5′ untranslated region (5′UTR) in apex-resected hearts, whereas there was an increase in coding regions (CDS) and stop codon regions compared with sham operated hearts (*Figure 8—figure supplement 1B*). We further investigated the m$^6$A distribution patterns in total and unique peaks. In contrast to total peaks which had an identical distribution, unique injury-dependent peaks demonstrated a distinct pattern in which a great increase in m$^6$A deposits appeared in the CDS together with a relative decrease in both 5′UTRs and non-coding RNAs (*Figure 8—figure supplement 1C*). As expected, an increase in total m$^6$A peaks and m$^6$A-tagged mRNAs was detected in injured hearts (*Figure 8—figure supplement 1D*). In a comparison of the sham-operated and apex-resected hearts, a total of 750 genes (635 upregulated

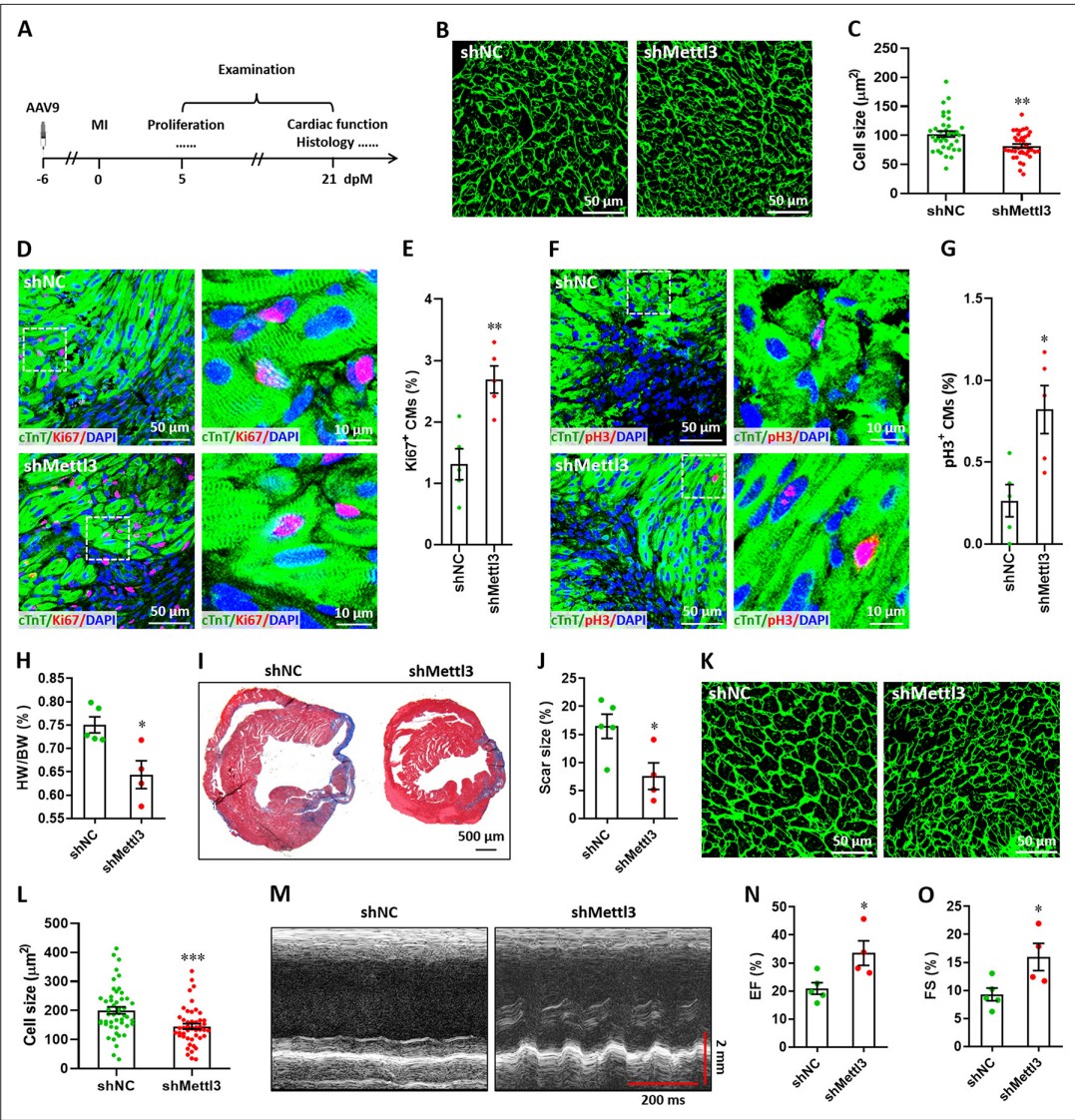

**Figure 6.** Mettl3 deficiency promotes heart regeneration in postnatal mice injured at p7. (**A**) Schematic of AAV9 virus injection at p1, myocardium infarction (MI) induction at p7, and histological analysis at 5 and 21 dpM. (**B and C**) Representative WGA staining images (**B**) and quantification of cardiomyocyte size (**C**) in control and Mettl3-deficient hearts at 5 dpM (n=~100 cells from 5 hearts). (**D and E**) Representative images (**D**) and quantification (**E**) of Ki67+ cardiomyocytes in the border zone of injured hearts at 5 dpM (n=5 hearts). (**F and G**) Representative images (**F**) and quantification (**G**) of pH3+ cardiomyocytes in the border zone of injured hearts at 5 dpM (n=5 hearts). (**H**) Quantification of heart weight (HW) to body weight (BW) ratio at 21 dpM (n=5 hearts). (**I and J**) Representative masson's trichrome staining images (**I**) and quantification of scar size (**J**) in control and Mettl3-deficient hearts at 21 dpM (n=5 hearts). (**K and L**) Representative WGA staining images (**K**) and quantification of cardiomyocyte size (**L**) in control and Mettl3-deficient hearts at 21 dpM (n=~100 cells from 5 hearts). (**M–O**) Representative images of M-model echocardiography (**M**) and quantification of LVEF (**N**) and LVFS (**O**) are shown at 21 dpM (n=5 hearts). All data are presented as the mean ± SEM, *p<0.05, **p<0.01, ***p<0.001 versus control. p values were determined by 2-tailed Student's t-test.

The online version of this article includes the following source data and figure supplement(s) for figure 6:

**Figure supplement 1.** Mettl3 expression in postnatal and adult hearts upon myocardium infarction injury.

**Figure supplement 1—source data 1.** Western blot for *Figure 6—figure supplement 1A* showing Mettl3 expression.

**Figure supplement 1—source data 2.** Western blot for *Figure 6—figure supplement 1A* showing β-actin expression.

*Figure 6 continued on next page*

*Figure 6 continued*

**Figure supplement 1—source data 3.** Western blot for *Figure 6—figure supplement 1B* showing Mettl3 expression.

**Figure supplement 1—source data 4.** Western blot for *Figure 6—figure supplement 1B* showing β-actin expression.

and 115 downregulated) were identified with a 2-fold m⁶A change (*Figure 8—figure supplement 1E*). Gene Ontology (GO) analysis of these 635 upregulated genes indicated that a handful of the genes were associated with cardiac formation and assembly, hippo signaling, and establishment of spindle orientation (*Figure 8—figure supplement 1F*).

In addition, both m⁶A-tagged transcripts overlapped between two different repeats and total MeRIP-seq reads were increased in the resected hearts compared with sham (*Figure 8—figure supplement 2A, B*), indicating an increase in m⁶A-tagged transcripts in injured hearts. A substantial proportion (about 60%) of total m⁶A-tagged transcripts (1,458 of 2,511) were specifically detected in the resected rather than sham hearts (*Figure 8—figure supplement 2C*), implying that heart injury results in different landscapes of m⁶A modification. Compared with the 1,053 overlapping

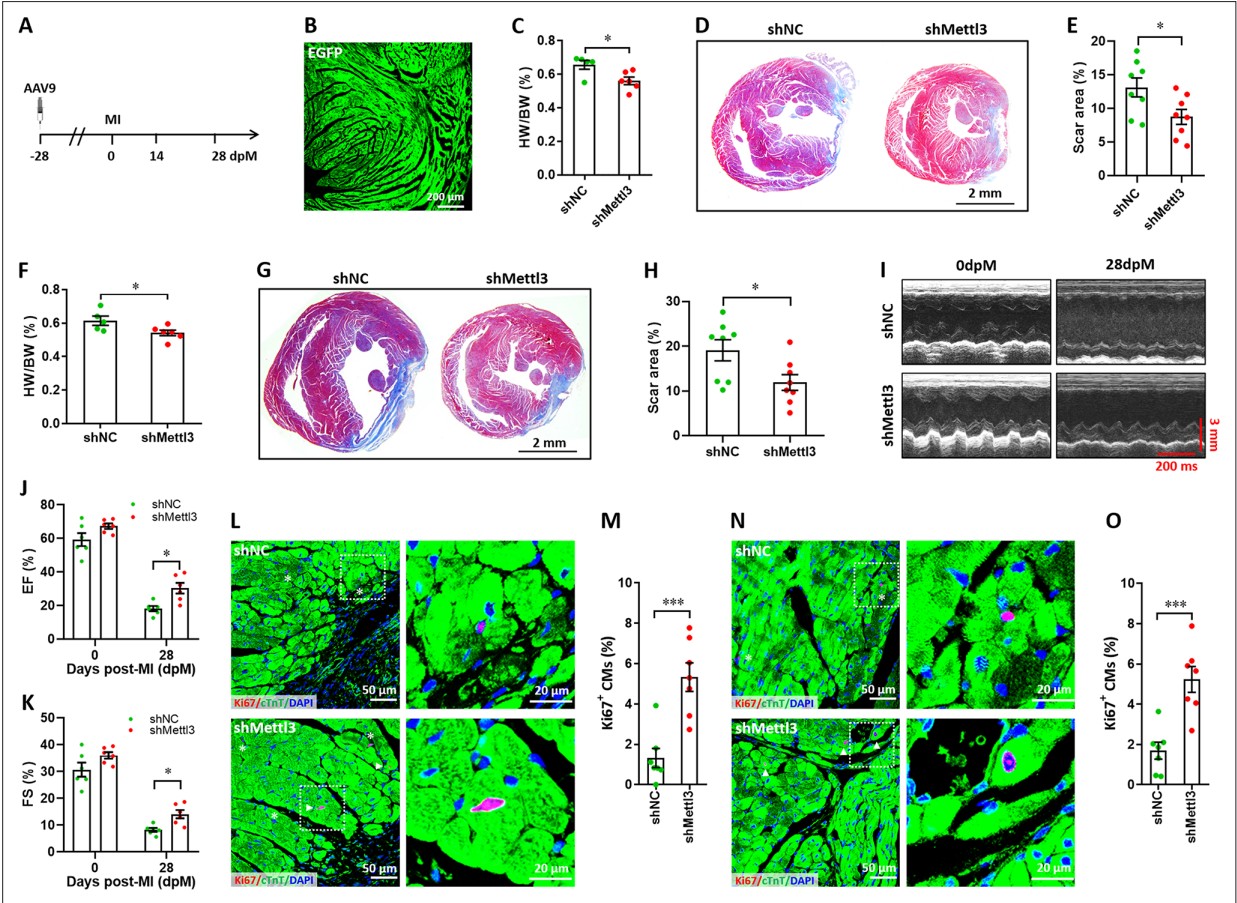

**Figure 7.** Mettl3 deficiency promotes heart regeneration in adult mice upon myocardium infarction injury. (**A**) Schematic of AAV9 virus injection 4 weeks prior to myocardium infarction (MI), followed by histological analysis at 14 and 28 dpM. (**B**) Representative images of reporter gene (EGFP) expression mediated by AAV9-sh*Mettl3* virus in heart at 0 dpM. (**C**) Quantification of heart weight (HW) to body weight (BW) ratio at 14 dpM (n=6 hearts). (**D and E**) Representative masson's trichrome staining images (**D**) and quantification of scar size (**E**) in control and Mettl3-deficient hearts at 14 dpM (n=8 hearts). (**F**) Quantification of HW to BW ratio at 28 dpM (n=6 hearts). (**G and H**) Representative masson's trichrome staining images (**G**) and quantification of scar size (**H**) in control and Mettl3-deficient hearts at 28 dpM (n=8 hearts). (**I–K**) Representative images of M-model echocardiography (**I**) and quantification of LVEF (**J**) and LVFS (**K**) are shown at 0 and 28 dpM (n=6 hearts). (**L and M**) Representative images (**L**) and quantification (**M**) of Ki67⁺ cardiomyocytes in the border zone of injured hearts at 14 dpM (n=7 hearts). (**N and O**) Representative images (**N**) and quantification (**O**) of Ki67⁺ cardiomyocytes in the remote zone of injured hearts at 14 dpM (n=7 hearts). All data are presented as the mean ± SEM, *p<0.05, ***p<0.001. *P* values were determined by 2-tailed Student's t-test (C, E, F, H, M, and O), or by 2-way ANOVA with Dunnett's multiple-comparison test (**J and K**).

transcripts, GO analysis revealed that there were different GO biological terms including cell cycle for the 1,458 transcripts specifically tagged by m⁶A in injured hearts (*Figure 8—figure supplement 2D, E*).

## Mettl3-mediated m⁶A modification downregulates Fgf16 expression in cardiomyocytes

RNA-seq analysis identified 3,213 upregulated and 809 downregulated genes at 5 dpr compared with sham group (*Figure 8—figure supplement 3A, B*). Gene-set enrichment analysis (GSEA) revealed that cardiac muscle cell development and proliferation gene sets are related to neonatal heart injury and regeneration (*Figure 8—figure supplement 3C, D*). Based on RNA-seq and m⁶A-seq data, 333 overlapped genes between the m⁶A-tagged and mRNA-upregulated genes were identified at 5 dpr (*Figure 8—figure supplement 4A*). GO analysis revealed that chromatin structure adjustment and cell cycle are important for cardiac regeneration (*Figure 8—figure supplement 4B, C*). 7 of 19 cell cycle-related genes and 5 of 13 proliferation-related genes were upregulated at least 2-fold (*Figure 8—figure supplement 4D, E*), implying that these cell cycle- and proliferation-related genes might play important roles in the regulation of heart regeneration in a Mettl3-m⁶A dependent manner. Among these genes, qPCR validation further confirmed that *Mis12*, *Fgf16*, and *Six5* were significantly upregulated during heart regeneration (*Figure 8—figure supplement 4F*). To further explore the effects of these three genes on the proliferation of cardiomyocytes, H9c2 cells transfected with si*RNAs* were subjected to flow cytometry. We found that the silencing of *Fgf16*, rather than *Mis12* and *Six5*, significantly induced G1-phase cell cycle arrest (*Figure 8—figure supplement 4G-J*). Importantly, *Fgf16* silencing also suppressed the proliferation of primary cardiomyocytes isolated from neonatal mice, which was demonstrated by double staining with Ki67 and cTnT (*Figure 8A and B*). Moreover, the decreased proliferation of primary cardiomyocytes induced by *Fgf16* silencing was further confirmed by nuclear EdU incorporation (*Figure 8C and D*). Consistent with previous reports (*Hotta et al., 2008*; *Yu et al., 2016*), our results imply that Fgf16 is vital for promoting cardiomyocyte proliferation in neonatal mice.

GSEA revealed that the Fgf signaling pathway is related to neonatal heart regeneration (*Figure 8E*). Importantly, MeRIP-seq showed a great increase (about 10 folds) in the m⁶A peaks of *Fgf16* mRNA during neonatal heart injury (*Figure 8F*, upper panel). However, the RNA-seq reads were only increased by 2 folds in the Fgf16 mRNA (*Figure 8F*, lower panel). These disproportionate increases indicate that the MeRIP-seq reads are true reads mapping to m⁶A modification and not reads captured due to increased abundance for Fgf16 mRNA. Therefore, these data suggest the involvement of the m⁶A modification on *Fgf16* mRNA during heart regeneration. To further determine whether and how m⁶A modification regulates *Fgf16* mRNA levels, we performed m⁶A-RIP-qPCR and found that m⁶A enrichment in *Fgf16* was remarkably upregulated by *Mettl3* overexpression (*Figure 8G*) but suppressed by *Mettl3* silencing (*Figure 8H*) in the regenerating heart at 5 dpr. However, *Fgf16* mRNA expression at 5 dpr was significantly suppressed by *Mettl3* overexpression (*Figure 8I*). As expected, *Mettl3* silencing significantly increased the mRNA levels of *Fgf16* (*Figure 8I*). To further explore the effects of Mettl3 on Fgf16 expression in neonatal heart, western blotting was used to check the protein levels of Fgf16. Our data showed that Mettl3 knockdown (*Figure 8J*) significantly increased the production of Fgf16 protein in the regenerating heart at 5 dpr (*Figure 8K*). Mettl3 knockdown also increased the protein levels of Fgf16 at p8 (equal to 5 dpr in the injured heart) in the uninjured neonatal heart, although the expression levels were lower than that in the injured heart at 5 dpr (*Figure 8L*). Therefore, these findings suggest that Mettl3-mediated m⁶A negatively regulates the post-transcriptional levels of *Fgf16* during heart injury.

To further determine the cell types primarily expressed and secreted FGF16 in the heart, we examined the expression pattern of Fgf16 inCMs and nCMs during heart regeneration. After injection of AAV9 viruses at p1, CMs and nCMs were isolated from hearts in neonatal mouse at 5 dpr. Isolated CMs and nCMs were directly subjected to protein extraction and western blotting assay. As shown in *Figure 8M*, Fgf16 expressions in CMs were greatly increased by Mettl3 deficiency compared with control hearts during regeneration. However, the expression of Fgf16 in nCMs was decreased in the Mettl3-deficient hearts compared with controls. Importantly, Fgf16 expression level in CMs is significantly higher than that in nCMs during heart regeneration regardless of Mettl3 deficiency or not (*Figure 8M*). Therefore, these findings indicate that the great changes in the m⁶A modification and

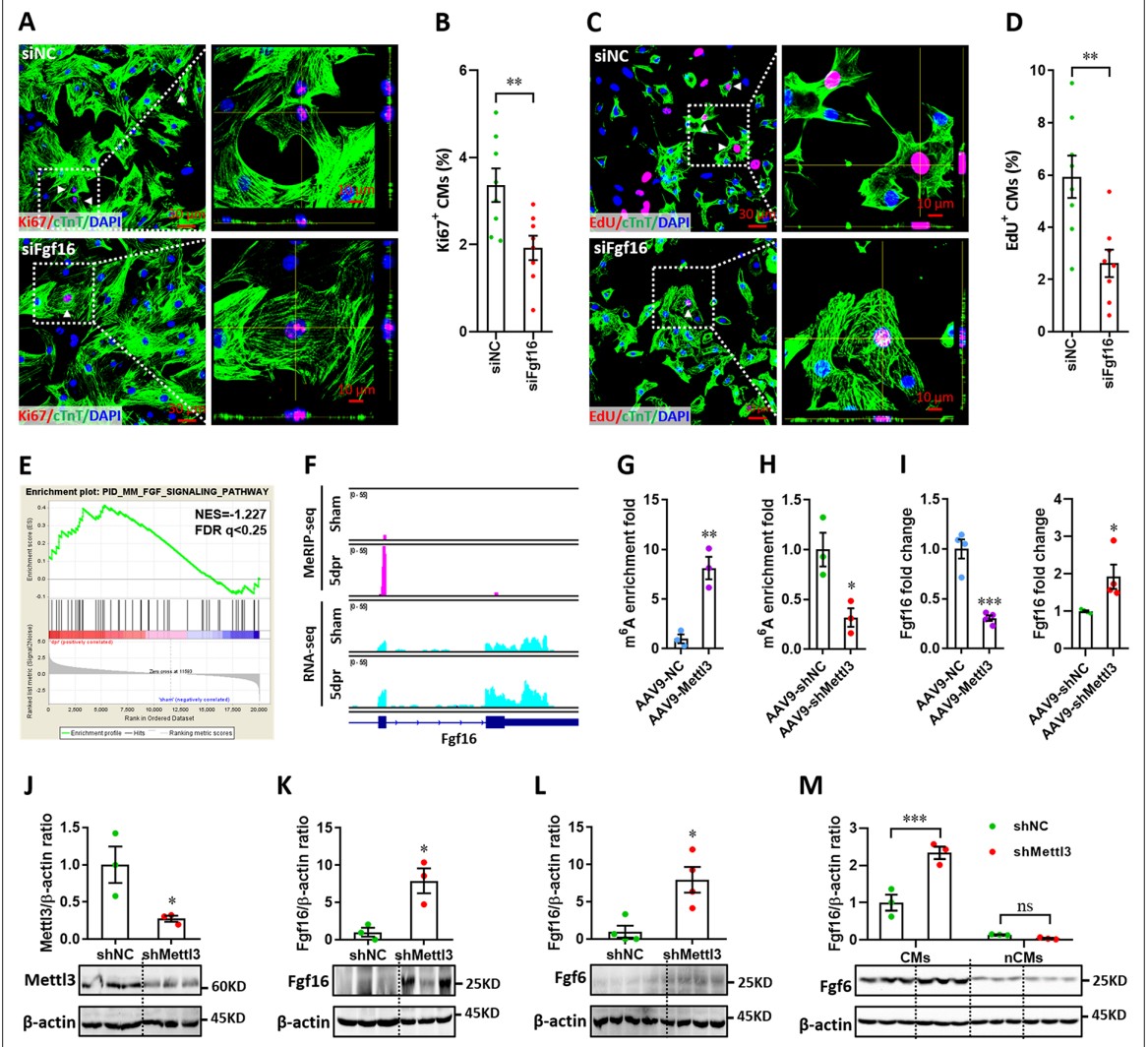

**Figure 8.** Mettl3-mediated m⁶A post-transcriptionally regulates Fgf16 during neonatal heart regeneration. (**A and B**) Primary cardiomyocytes were transfected with si*Fgf16* and si*NC* for 48 hr, followed by Ki67 (red) and cTnT (green) double staining. Representative images (**A**) and quantification (**B**) of Ki67⁺ cardiomyocytes are shown (n=8). Right panel (**A**), magnified Z-stack confocal images. (**C and D**) Primary cardiomyocytes were transfected with si*Fgf16* and si*NC* for 48 hr, followed by EdU incorporation assay. Representative images (**C**) and quantification (**D**) of EdU⁺ cardiomyocytes are shown (n=8). Right panel (**C**), magnified Z-stack confocal images. (**E**) GSEA analysis indicated the involvement of Fgf pathway in neonatal heart regeneration. (**F**) Representative IGV plots of mouse MeRIP-seq (upper panel) and RNA-seq (lower panel) reads for *Fgf16*. (**G and H**) MeRIP-qPCR quantification of m⁶A enrichment in *Fgf16* mRNA in Mettl3-overexpressing (**G**) and Mettl3-silencing (**H**) hearts at 5 dpr (n=3 hearts). (**I**) qPCR validation of *Fgf16* in the injured hearts with AAV9 virus injection at 5 dpr (n=4 hearts). (**J and K**) AAV9-sh*Mettl3* viruses were injected into neonatal mice at p1, followed by apex resection at p3 and samples collection at 5dpr. Mettl3 knockdown (J, n=3 hearts) and Fgf16 expression (K, n=3 hearts) in neonatal hearts at 5 dpr were then validated by western blotting. (**L**) Validation of Fgf16 protein expression in homeostatic neonatal hearts at p8 after AAV9-sh*Mettl3* injection at p1 (n=4 hearts). (**M**) Neonatal mice were injected with AAV9-sh*Mettl3* virus at p1 and resected at p3. Fgf16 expression in primary cardiomyocytes (CMs) and non-cardiomyocytes (nCMs) isolated at 5 dpr were then validated by western blotting (n=3). All data are presented as the mean ± SEM, *p<0.05, **p<0.01, ***p<0.001 versus control. p values were determined by 2-tailed Student's t-test (**A–L**), or by 2-way ANOVA with Dunnett's multiple-comparison test (**M**).

The online version of this article includes the following source data and figure supplement(s) for figure 8:

**Source data 1.** Western blot for *Figure 8J* showing Mettl3 expression.

**Source data 2.** Western blot for *Figure 8J* showing β-actin expression.

**Source data 3.** Western blot for *Figure 8K* showing Fgf16 expression.

**Source data 4.** Western blot for *Figure 8K* showing β-actin expression.

**Source data 5.** Western blot for *Figure 8L* showing Fgf16 expression.

*Figure 8 continued on next page*

*Figure 8 continued*

**Source data 6.** Western blot for *Figure 8L* showing β-actin expression.

**Source data 7.** Western blot for *Figure 8M* showing Fgf16 expression.

**Source data 8.** Western blot for *Figure 8M* showing β-actin expression.

**Figure supplement 1.** Variations of m⁶A-tagged transcripts in neonatal heart in response to injury.

**Figure supplement 2.** Apex resection upregulates levels of m⁶A-tagged transcripts in neonatal hearts.

**Figure supplement 3.** PCA, volcano plot, and GSEA analysis from RNA-seq data.

**Figure supplement 4.** Identification of cell cycle and proliferation-related target genes from the overlapped gene sets between m⁶A-tagged and RNA-upregulated genes.

**Figure supplement 5.** Luciferase reporter assay for Mettl3-mediated Fgf16 expression.

**Figure supplement 6.** Effects of Fgf16 knockdown on Mettl3-mediated regulation of CM proliferation and heart regeneration.

**Figure supplement 7.** Effects of Mettl14 and Fto on Fgf16 expression and cell proliferation in CMs.

**Figure supplement 8.** Mettl3 regulates Fgf16 expression in an Ythdf2-dependent manner.

expression of Fgf16 in the Mettl3-deficient hearts during regeneration dominantly results from CMs rather than nCMs.

To further address the effects of m⁶A modification on *Fgf16* expression, we constructed reporter genes by fusing luciferase to wild-type Fgf16 coding region with the m⁶A motif (GGACU) identified by MeRIP-seq (*Figure 8F*) or to mutant Fgf16 (ΔFgf16) in which the m⁶A motif was replaced by GGGCU (*Figure 8—figure supplement 5A*). The relative luciferase activity and mRNA level of the wild-type and mutant Fgf16-fused reporter genes were compared in H9c2 cells. For the wild-type reporter gene, both the luciferase activity and mRNA levels were greatly suppressed by *Mettl3* overexpression, and this suppression was significantly blocked by Mettl3 catalytic mutant (ΔMettl3) and the mutation of m⁶A consensus sequence in *Fgf16* (ΔFgf16), respectively (*Figure 8—figure supplement 5B, C*). Taken together, these findings suggest that the levels of *Fgf16* mRNA is negatively controlled by Mettl3-mediated m⁶A modification.

## Fgf16 is a direct target of Mettl3-mediated heart regeneration

To further evidence the direct role of Fgf16 in Mettl3-mediated regulation of CM proliferation and heart regeneration, we further assessed the impact of Fgf16 knockdown on Mettl3-mediated regulation of CM proliferation and heart regeneration. First, primary CMs isolated from neonatal mice were treated with siRNA control (si*NC*), si*Mettl3*, and si*Mettl3* +si*Fgf16* for 48 hr, respectively. Ki67 and cTnT double staining revealed that siMettl3 treatment greatly increased the percentage of Ki67⁺ cTnT⁺ cells compared with control group, indicating an increased proliferation level of CMs. However, si*Mettl3*-increased CM proliferation was significantly blocked by siFgf16 treatment (*Figure 8—figure supplement 6A, B*). These in vitro data suggest that suppressing Fgf16 expression in Mettl3-deficient CMs can greatly block its proliferation. To further confirm this idea, we also injected neonatal mice at p1 by AAV9-sh*NC*, AAV9-sh*Mettl3*, and AAV9-sh*Mettl3*+AAV9-sh*Fgf16*, respectively. Apex resection was then performed at p3. Heart tissues were collected at 5 dpr and 14 dpr for analysis, respectively (*Figure 8—figure supplement 6C*). In agreement with in vitro data, immunofluorescent staining revealed that sh*Mettl3* injection increased CM proliferation in ventricular apex at 5 dpr compared with control groups. However, the sh*Mettl3*-increased CM proliferation was significantly blocked by sh*Fgf16* injection (*Figure 8—figure supplement 6D, E*). In addition, histological analysis revealed that scar size in the cardiac apex at 14 dpr was greatly decreased by Mettl3 deficiency compared with control groups. However, this sh*Mettl3*-induced decrease in scar size was significantly blocked by Fgf16 knockdown (*Figure 8—figure supplement 6F, G*). In consistent, cardiac function analysis revealed that sh*Mettl3*-inceased left ventricular systolic function was significantly blocked by sh*Fgf16* (*Figure 8—figure supplement 6H-J*). Taken together, our data strongly evidenced the important role of Fgf16 in Mettl3-mediated regulation of CM proliferation and heart regeneration.

To determine whether another m⁶A regulatory protein has a similar impact on Fgf16 expression and therefore on cardiomyocyte proliferation, Mettl14 and Fto were further examined under our experimental conditions. Primary CMs isolated from neonatal mice at p1 were treated with si*Mettl14* and si*Fto* for 48 hr, followed by qPCR assay and immunofluorescent staining assay. Our data showed that

Mettl14 and Fto knockdown (*Figure 8—figure supplement 7A, B*) did not significantly change the expression level of Fgf16 in primary cardiomyocytes (*Figure 8—figure supplement 7C*). Moreover, the percentage of Ki67-positive cardiomyocytes in si*Mettl14*- and si*Fto*-treated groups was comparable with control group (*Figure 8—figure supplement 7D, E*). These data suggest that Mettl14 and Fto knockdown has no significant effects on Fgf16 expression and cardiomyocyte proliferation. Taking these findings together, it is reasonable that m⁶A-mediated Fgf16 expression, cardiomyocyte proliferation, and heart regeneration depend on Mettl3 rather than other m⁶A regulators.

## Ythdf2 is involved in Mettl3-mediated downregulation of Fgf16

It has been demonstrated that the m⁶A methylation sites can be recognized by 'readers', and therefore impacting the fate of the target mRNA (*Shi et al., 2019*). Among the popular m⁶A 'readers', cytoplasmic Ythdf2 facilitates decay of its target transcripts with m⁶A modification under normal and stress conditions (*Wang et al., 2014a*; *Du et al., 2016*; *Ivanova et al., 2017*). By contrast, Ythdf1, the other m⁶A reader, promotes translation of target transcripts (*Wang et al., 2015*; *Shi et al., 2018*; *Shi et al., 2019*). To determine the potential involvement of these two representative m⁶A readers in Fgf16 regulation, we transfected primary cardiomyocytes with si*RNA*s and examined the expression of *Fgf16*. Silencing of *Ythdf2* rather than *Ythdf1* elevated the expression of *Fgf16* up to 6-fold in primary cardiomyocytes (*Figure 8—figure supplement 8A and B*). Moreover, we performed RNA decay assay and found that relative *Fgf16* mRNA level was increased in *Ythdf2*-deficient cardiomyocytes in comparison to control cells after Actinomycin-D treatment for 3–6 hr (*Figure 8—figure supplement 8C*), indicating the decreased degradation of *Fgf16* mRNA. These data suggest that Ythdf2 may induce decay of *Fgf16* mRNA tagged by m⁶A, thereby negatively regulating *Fgf16* expression. In order to verify whether Ythdf2 participates in m⁶A modification of *Fgf16* mRNA, RIP-qPCR was used to investigate the interaction between *Fgf16* mRNA and Ythdf2. Our data showed that Ythdf2 remarkably enriched in *Fgf16* mRNA in normal primary cardiomyocytes, whereas this relative enrichment was significantly suppressed in *Mettl3*-deficient cardiomyocytes (*Figure 8—figure supplement 8D*).

To further elucidate whether Ythdf2-regulated expression of *Fgf16* depends on m⁶A modification, wild-type and *Ythdf2*-deficient (*Figure 8—figure supplement 8E*) H9c2 cells were co-transfected with wild-type or mutant Fgf16 reporter genes, respectively. Our data showed that the relative luciferase activity of wild-type reporter gene was greatly increased by *Ythdf2* knockdown. For mutant Fgf16 reporter plasmid, *Ythdf2* knockdown did not alter the relative luciferase activity. However, the mutant Fgf16 reporter plasmid showed increased luciferase activity in normal cells (si*NC*) compared with the wild-type reporter plasmid (*Figure 8—figure supplement 8F*). These results suggest that Ythdf2 is involved in the Mettl3-mediated downregulation of Fgf16 by inducing decay of *Fgf16* mRNA.

## Effects of Fgf16 mutant on neonatal heart regeneration

Finally, we investigated the effects of Fgf16 mutant (ΔFgf16) on neonatal heart regeneration. To overexpress wild-type and mutant Fgf16 in cardiomyocytes, AAV9-Fgf16 and AAV9-ΔFgf16 viruses (*Figure 9A*) were injected into neonatal mice at p1, followed by apex resection at p3 and sample collection at the indicated time points (*Figure 9B*). AAV9-NC virus was used as negative control. The expression of target gene was determined by qPCR. Our data showed that the protein level of Fgf16 in neonatal heart was slightly elevated by AAV9-Fgf16 virus. However, AAV9-ΔFgf16 remarkably increased the expression of Fgf16 protein compared with AAV9-Fgf16 group (*Figure 9C*). To determine whether ΔFgf16 influences cardiomyocyte proliferation during heart regeneration, immunofluorescent staining was performed in cardiac tissues at 5 dpr using different proliferating markers including Ki67, pH3, and Aurora B. Ki67 staining showed that there was no significant difference in the proliferation of cardiomyocytes in cardiac apex in AAV9-Fgf16 group in comparison to AAV9-NC group. However, AAV9-ΔFgf16 significantly increased the percentage of Ki67⁺ cTnT⁺ cells compared with AAV9-Fgf16 group (*Figure 9D and E*). The percentage of pH3⁺ cTnT⁺ cells was also significantly elevated by AAV9-ΔFgf16 compared with AAV9-Fgf16, although AAV9-Fgf16 only slightly promoted the percentage of pH3⁺ cTnT⁺ cells in comparison to AAV9-NC (*Figure 9F and G*). Moreover, increased percentage of AurkB⁺ cardiomyocytes was also detected in cardiac apex in AAV9-ΔFgf16 group compared with AAV9-Fgf16 group (*Figure 9H1*). These findings strongly revealed that the mutation of m⁶A consensus sequence in *Fgf16* (ΔFgf16) can attenuate m⁶A/Ythdf2-mediated RNA decay during heart regeneration, thereby promoting cardiomyocyte proliferation. As expected, trichrome staining

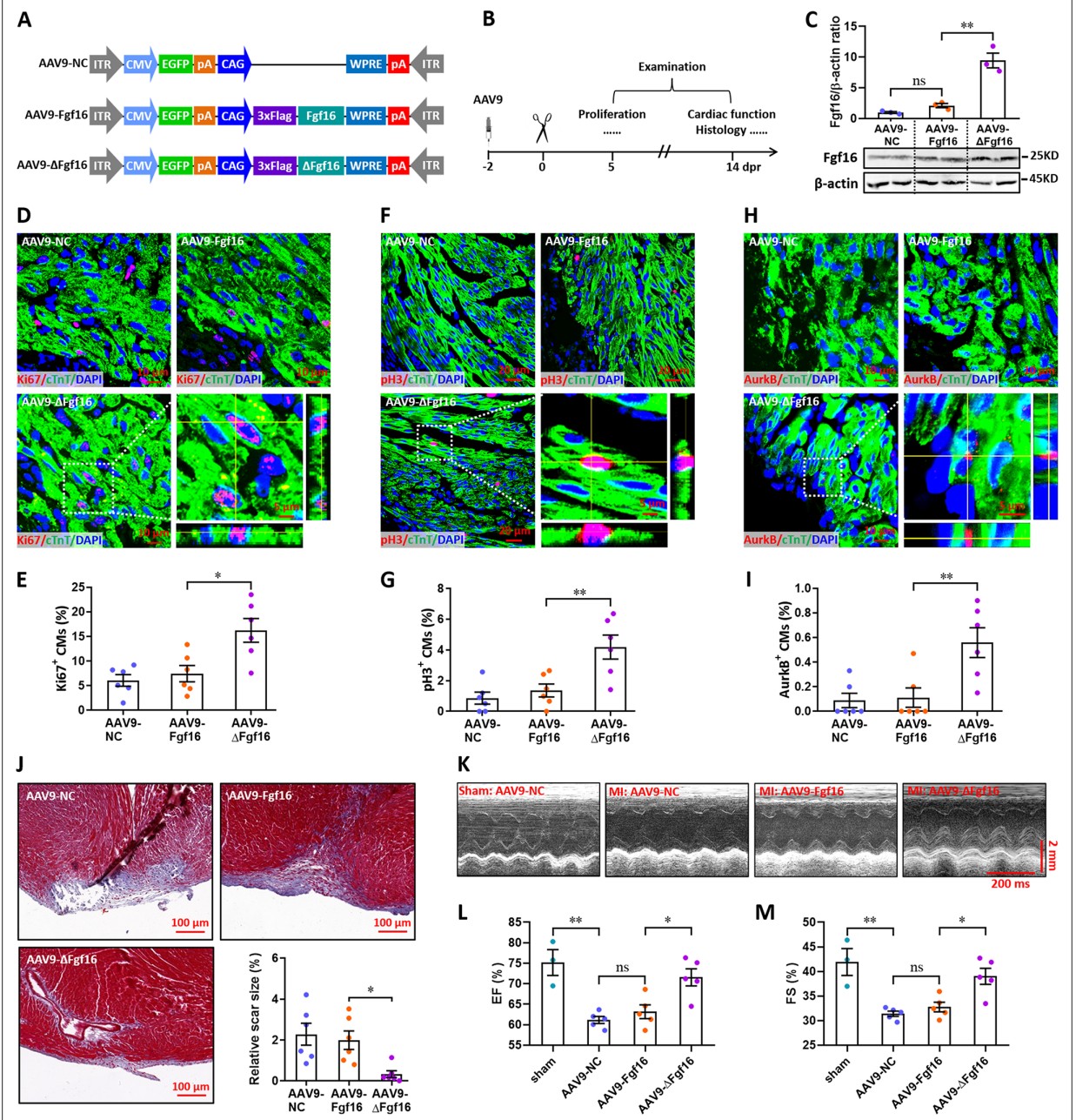

**Figure 9.** Mutation of the consensus sequence of m⁶A modification in *Fgf16* mRNA promotes heart regeneration. (**A**) Schematic of AAV9 vectors for the expression of negative control (AAV9-NC), wild-type Fgf16 (AAV9-Fgf16), and mutant Fgf16 (AAV9-ΔFgf16). (**B**) Schematic of AAV9 virus injection, apex resection, and sample collection in neonatal mice. (**C**) Representative images (lower panel) and quantification (upper panel) of Western blotting validation of Fgf16 protein expression in neonatal hearts at 5 dpr (n=3 hearts). (**D and E**) Representative images (**D**) and quantification (**E**) of Ki67⁺ cardiomyocytes in apical ventricle at 5 dpr (n=6 hearts). Right lower panel (**D**), magnified Z-stack confocal image of Ki67⁺ cardiomyocyte. (**F and G**) Representative images (**F**) and quantification (**G**) of pH3⁺ cardiomyocytes in apical ventricle at 5 dpr (n=6 hearts). Right lower panel (**F**), magnified Z-stack confocal image of pH3⁺ cardiomyocyte. (**H and I**) Representative images (**H**) and quantification (**I**) of AurkB⁺ cardiomyocytes in apical ventricle at 5 dpr (n=6 hearts). Right lower panel (**H**), magnified Z-stack confocal image of AurkB⁺ cardiomyocyte. (**J**) Representative masson's trichrome staining images of cardiac apex and quantification of scar size in hearts at 14 dpr (n=6 hearts). (**K–M**) Representative images of M-model echocardiography (**K**) and quantification of LVEF (**L**) and LVFS (**M**) are shown at 14 dpr (n=3 mice for sham, and 5 mice for each injured group). All data are presented as the mean ± SEM, *p<0.05, **p<0.01 versus control, ns means no significant difference. p values were determined by 1-way ANOVA with Dunnett's multiple-comparison test.

The online version of this article includes the following source data for figure 9:

**Source data 1.** Western blot for *Figure 9C* showing Fgf16 expression.

**Source data 2.** Western blot for *Figure 9C* showing β-actin expression.

results further revealed a decreased fibrotic scar size at 14 dpr in AAV9-ΔFgf16 group compared with AAV9-Fgf16 group (*Figure 9J*), indicating the accelerated heart regeneration. Consistent with these results, cardiac function analysis revealed that there was a higher left ventricular systolic function in AAV9-ΔFgf16 group compared with AAV9-Fgf16 group (*Figure 9K–M*). Taken together, these results suggest that the mutation of m⁶A consensus sequence in *Fgf16* (ΔFgf16) can accelerate postnatal heart regeneration by promoting cardiomyocyte proliferation.

## Discussion

In this study, we provide compelling in vitro and in vivo evidence demonstrating the important role of Mettl3 in cardiomyocyte proliferation and heart regeneration in mice upon injury. We identified Mettl3 as a key myocardial methylase that regulates cardiac m⁶A and provided a novel characterization of Mettl3-mediated m⁶A modification during heart regeneration in mice. Mettl3 deficiency increases cardiomyocyte proliferation in the early stage of injury and accelerates heart regeneration in post-natal mice (*Figures 2–4*). However, overexpression of Mettl3 attenuated cardiomyocyte proliferation, heart regeneration, and cardiac function (*Figure 5*). Our findings reveal that the Mettl3-mediated m⁶A modification of myocardial mRNAs plays an important role in regulating neonatal heart regen-eration, demonstrating novel therapeutic potential for Mettl3-mediated m⁶A modification in heart regeneration.

Mechanistically, our work demonstrates a critical role for m⁶A modification in the regulation of mammalian heart regeneration upon injury. Indeed, several studies have recognized the importance of m⁶A modification in tissue and/or cell regeneration. The sciatic nerve injury elevates the levels of m⁶A-tagged transcripts encoding many regenerative genes and protein translation machinery components in adult mouse dorsal root ganglion, indicating a critical epitranscriptomic mechanism for promoting injury-induced axon regeneration in the adult mammalian nervous system (*Weng et al., 2018*). Moreover, expression of *Ythdf2* in hematopoietic stem cells (HSCs) facilitates the decay of the m⁶A-modified Wnt target gene mRNAs, thereby suppressing the Wnt signaling pathway. However, deletion of *Ythdf2* expands the number of HSCs and increased the regenerative capacity of HSCs under stress conditions, suggesting that Ythdf2-mediated m⁶A-dependent mRNA clearance facilitates hematopoietic stem cell regeneration (*Wang et al., 2018b*). A recent study revealed that Fto, an m⁶A demethylase, plays a critical role in cardiac contractile function during homeostasis, remodeling, and regeneration in an m⁶A-dependent manner (*Mathiyalagan et al., 2019*). These findings are consistent with our data and reveal that m⁶A modification is vital for the regulation of tissue regeneration.

m⁶A is added to mRNAs by a methyltransferase complex including Mettl3, Mettl14, and Wilm's tumor 1-associating protein (Wtap), and among these proteins, Mettl3 was identified to function as a catalytic subunit (*Bokar et al., 1997*; *Liu et al., 2014*). In mammals, Mettl3 affects cell division, differentiation, reprogramming, and embryonic development. Knockout of *Mettl3* in mouse embry-onic stem cells impairs differentiation and *Mettl3⁻ᐟ⁻* mice are embryonic lethal (*Geula et al., 2015*). Consistent with these data, previous studies have demonstrated that Mettl3-mediated m⁶A modifica-tion is critical for cerebellar development (*Wang et al., 2018a*), HSC differentiation (*Vu et al., 2017*), and acute myeloid leukemia maintenance (*Barbieri et al., 2017*). These previous studies suggest an important role of Mettl3-mediated m⁶A mRNA methylation in physiology and pathology. In this study, *Mettl3* knockdown promoted cardiomyocyte proliferation and accelerated heart regeneration in post-natal mice in an m⁶A-dependent manner, whereas overexpression of *Mettl3* was sufficient to suppress cardiomyocyte proliferation, heart regeneration and cardiac function (*Figures 2–5*). Our data suggest that Mettl3-mediated m⁶A negatively regulates heart regeneration in neonatal mice. Consistent with our data in neonatal mice (*Figures 4 and 5*), previous study reported that *Mettl3* overexpression promotes cardiomyocyte hypertrophy in adult mice, but deficiency in *Mettl3* abolishes the ability of cardiomyocytes to undergo hypertrophy in response to stimulation (*Dorn et al., 2019*). Additionally, controversial functions of m⁶A modification were also detected in the differentiation and reprogram-ming of stem cells. For instance, *Mettl* knockdown reduced self-renewal abilities in mouse embryonic stem cells (*Wang et al., 2014b*). However, a later study revealed a crucial role for *Mettl3* in embryonic priming from naïve pluripotency toward a more differentiated lineage (*Geula et al., 2015*). A reason-able explanation for the different functions of Mettl3 might be attributed to its multifaceted cellular localization, stability, and translational modification under different conditions (*Shi et al., 2019*). In contrast with injured animal model, our data showed that Mettl3 gain and loss of function does not

significantly influence cardiomyocyte proliferation and cardiac function in the homeostatic postnatal mice without injury stresses (*Figure 3—figure supplement 3*, *Figure 4—figure supplement 1*, and *Figure 5—figure supplement 3*). The different effects of Mettl3 between homeostatic and injured hearts might result from the robust changes in m$^6$A modification during heart injury and regeneration. In agreement with this hypothesis, great increases in the levels of m$^6$A modification and Mettl3 expression were indeed observed in the regenerating neonatal hearts at 5 dpr compared with sham hearts (*Figure 1*).

Using RNA-seq and MeRIP-seq, we identified Fgf16 as one of downstream targets of Mettl3-mediated m$^6$A modification. This finding was further verified by MeRIP-qPCR and luciferase reporter gene assays. Mechanistically, our data demonstrate that Mettl3 epigenetically silences *Fgf16* in cardiomyocytes through an m$^6$A-Ythdf2-dependent mechanism (*Figure 8*, *Figure 8—figure supplement 5*, and *Figure 8—figure supplement 8*). Fgf16 belongs to the Fgf9 subfamily and is the only cardiac-specific Fgf family member. A previous study has reported the predominant expression of Fgf16 in cardiomyocytes and a significant decrease in the proliferation of embryonic cardiomyocytes in *Fgf16$^{-/-}$* mice (*Hotta et al., 2008*). Consistently, Fgf16 has been demonstrated to be required for embryonic heart development and cardiomyocyte replication (*Lu et al., 2008*). In addition, a recent study has shown that cardiac-specific overexpression of Fgf16 via AAV9 in *Gata4*-deficient hearts rescued cryoinjury-induced cardiac hypertrophy, promoted cardiomyocyte replication and improved heart function after injury (*Yu et al., 2016*). In line with these previous studies, our results show that *Fgf16* knockdown leads to a significant decrease in the proliferation levels of primary cardiomyocytes isolated from neonatal mice (*Figure 8*). In addition, the overexpression of mutant Fgf16 (mutation of m$^6$A consensus sequence) can significantly accelerate postnatal heart regeneration by promoting cardiomyocyte proliferation compared with wild-type Fgf16 (*Figure 9*). Given that there is a higher level of m$^6$A modification during heart regeneration (*Figure 1*), it is reasonable that wild-type *Fgf16* mRNA maintains a relative lower level compared with mutant *Fgf16*, due to the m$^6$A/Ythdf2-induced mRNA decay.

In the present study, increased levels of Mettl3, m$^6$A, and Fgf16 were observed in the injured neonatal heart at 5 dpr. It seems that there is a positive correlation between Mettl3 expression, m$^6$A levels, and Fgf16 expression. However, our data demonstrated that Mettl3-mediated m$^6$A negatively regulates the Fgf16 expression during heart injury (*Figure 8G–M*). Moreover, Mettl3 deficiency-increased cardiomyocyte proliferation and heart regeneration could be significantly blocked by Fgf16 silencing during heart regeneration (*Figure 8—figure supplement 6*), indicating the important role of Fgf16 in Mettl3-mediated regulation of heart regeneration. Given that the cardiac regenerative capacity of neonatal mouse only appears within one week after birth, we speculate that Fgf16 expression in neonatal heart is increased upon injury, thereby promoting heart regeneration. Meanwhile, the increased Mettl3 expression and m$^6$A modification in the injured neonatal heart may be responsible for preventing excessive expression of Fgf16 through Ythdf2-mediated RNA decay, thereby balancing cardiac regenerative capacity.

It is well known that heart is a heterogeneous organ and consists of several different cardiac cell types including cardiomyocytes, fibroblasts, endothelial cells and others. Among them, cardiomyocyte is the largest cell type in whole heart and has the highest percentage in all heart cells. Particularly, ventricular regions contain 49.2% cardiomyocytes, 21.2% mural cells, 15.5% fibroblasts, 7.8% endothelial cells and 5.3% immune cells (*Litviňuková et al., 2020*). In the present study, ventricular regions rather than entire hearts were used for RNA-seq and MeRIP-seq analysis. Therefore, it is reasonable that the greatly changed genes identified by RNA-seq and changed m$^6$A modification identified by MeRIP-seq during heart regeneration are likely to mainly result from cardiomyocytes. In line with this idea, the expression of Fgf16, the downstream target of Mettl3-mediated m$^6$A modification identified by the conjoint analysis of MeRIP-seq and RNA-seq, was dominantly expressed in CMs rather than nCMs regardless of injury stimulation (*Figure 8M*).

Although the m$^6$A modification is a part of the larger field of RNA epigenetics, yet its function during heart regeneration remains elusive. In this study, we demonstrate that Mettl3 post-transcriptionally reduces *Fgf16* mRNA levels through an m$^6$A-Ythdf2-depenten pathway, thereby restricting cardiomyocyte proliferation during heart regeneration in mice. Our data exemplify the pivotal role of m$^6$A epitranscriptomic changes in heart regeneration. Targeting the Mettl3-mediated m$^6$A modification of Fgf16 may represent a promising therapeutic strategy for promoting the proliferation of

cardiomyocytes in mammals. Our data on m6A-modification in hearts further our understanding of the mechanism of heart regeneration and provide innovative therapeutic interventions.

# Materials and methods

## Cell culture and treatment

H9c2 cells (rat cardiomyocyte cell line) were purchased from FuDan IBS Cell Center (FDCC, Shanghai, China) and were identified through STR profiling. Cells were cultured in high glucose DMEM (Corning, USA) with 10% fetal bovine serum (Gibco, USA), l μg/mL streptomycin (Gibco, USA), and 1 U/mL penicillin streptomycin (Gibco, USA), at 37°C in a 5% $CO_2$ incubator. Culture medium was replaced every 2–3 days and cells were passaged when they reached 80% confluence. Cells were mycoplasma negative through treatment with LookOut Mycoplasma Elimination Kit (Sigma-Aldrich). Primary cardiomyocytes were isolated from the hearts of neonatal mice as previously described (*Wu et al., 2021*). To knock down or overexpress target genes in cardiomyocytes, several methods, including siRNAs, lentivirus, and adeno-associated virus 9, were used as described below.

## Knockdown and overexpression of target genes in H9c2 cells

For Mettl3 knockdown in H9c2 cells, the specific si*RNA* for Mettl3 (si*Mettl3*) and negative control (si*NC*) were synthesized by RIBOBIO (Guangzhou, China). Cells were transfected with siMettl3 to knock down the expression of Mettl3. The knockdown of other target genes was also induced by si*RNA* transfection. Gene silencing was achieved by transfection of predesigned si*RNA* duplexes (*Supplementary file 1*) designed and synthesized by RIBOBIO (Guangzhou, China).

The coding sequence of Mettl3 was amplified from pcDNA3/Flag-METTL3 (Addgene, #53739) and inserted into the lentiviral plasmid pLOXCMV (*Qi et al., 2015*) to generate pLOX-Mettl3 overexpression plasmid. Viral particles were packaged in HEK 293T cells and used to infect H9c2 cells as previously described (*Qi et al., 2015*). The infected H9c2 cells were selected by puromycin and expanded to form a stable sub-line.

## Adeno-associated virus 9 (AAV9) production

The full-length Mettl3 coding sequences were amplified from the pcDNA3-Mettl3 plasmid (Addgene, #53739), and cloned into AAV serotype-9 expressing plasmid to overexpress Mettl3 (AAV9-Mettl3). AAV9-NC (virus packaged with empty plasmid) served as control for AAV9-Mettl3. In addition, Mettl3 specific shRNA or negative control (RiboBio, Guangzhou, China) were cloned into AAV9-sh*RNA* expressing plasmid to generate AAV9-sh*Mettl3* or AAV9-sh*NC* plasmids, respectively. To overexpress wild-type Fgf16 and mutant Fgf16 (ΔFgf16) in which the adenosine bases (GGACT) in m6A consensus sequences were replaced by guanine (GGGCT), both wild-type and mutant Fgf16 coding sequences were constructed into AAV9 plasmids for AAV9 virus generation. AAV9 viruses were packaged and produced using the AAV Helper-Free System (DongBio.Co.Ltd, Shenzhen, China). For AAV9 virus delivery in vivo, viruses were subcutaneously injected into neonatal mice at a dose of $1\times10^{11}$ V.G./mouse at postnatal day 1 (p1). The schematic of AAV9 virus injection can be found in related Figures and Figures supplement. Primary cardiomyocytes were infected with AAV9 virus at a dose of $5\times10^{10}$ V.G./well (24 well plate).

## Mettl3 knockdown and overexpression in primary cardiomyocytes

The isolation of primary cardiomyocytes was performed as previous described (*Nakada et al., 2017*). Briefly, fresh neonatal mice (within 2 days after birth) hearts were harvested and immediately fixed in 4% PFA/PBS at 4 °C for 4 hr. Hearts were subsequently incubated with collagenase IV (2.4 mg/ml, Sigma) and II (1.8 mg/ml, Sigma) for 12 hr at 37 °C. The supernatant was collected and spun down at 500 rpm for 2 min to yield the cardiomyocytes. The hearts were then minced to smaller pieces and the procedure was repeated until no more cardiomyocytes were dissociated from the hearts. Knockdown and overexpression of target genes in primary cardiomyocytes were induced by si*RNAs* transfection and AAV9-mediated expression, respectively. Sequences of si*RNAs* used in this study are listed in *Supplementary file 1*.

## In vitro cell proliferation assay

For cell proliferation assay, both primary cardiomyocytes and H9c2 cell line were incubated in 24 well plates with different treatments. DNA synthesis was then analyzed by 5-Ethynyl-2'-deoxyuridine (EdU)

labeling, using Cell-Light EdU Apollo567 In Vitro Imaging Kit (RiboBio, Guangzhou, China) according to the manufacturer's instructions. At least 7 images were randomly taken per well using a Zeiss LSM 700 laser confocal microscope (Carl Zeiss). The population of EdU⁺ cells was determined by counting at least 500 cells per well. The EdU⁺ cells were quantified as the percentage of total cells. Moreover, proliferation of primary cardiomyocytes was also performed by Ki67 (Abcam, ab16667, 1:250) staining. In addition, cell numbers were analyzed using the Enhanced Cell Counting Kit-8 (CCK-8, Beyotime Biotechnology, China) according to the manufacturer's instructions, as previously described (*Zhou et al., 2018*).

## Animals and heart injury models

All animal studies were approved by the Institutional Animal Care and Use Committee of Jinan University (IACUC-2018921–03) and conducted in accordance with the ARRIVE guidelines (*Percie du Sert et al., 2020*). In vivo experiments and animal management procedures were performed according to the NIH Guide for the Care and Use of Laboratory Animals. Both male and female C57BL/6 J mice were used for neonatal heart injury experiments. Apical resection surgeries were performed on neonatal mice at postnatal day 3 (p3) and p7, respectively, as described previously (*Porrello et al., 2011*). Hearts with or without EdU injection were collected at the indicated time points for further analysis. For in vivo knockdown or overexpression of target genes, AAV9 viruses were injected at p1, followed by apex resection (at p3 and p7) and sample collection at the indicated time points. In addition, myocardium infarction (MI) injury in adult male mice (8 weeks) was induced by permanent ligation of the left anterior descending artery (LAD) as previously described (*Wu et al., 2021*). In cases when anesthesia was required, mice were anaesthetized by oxygen and 2% isoflurane as previously reported (*Scott et al., 2021*). Animals were euthanized by dissecting the diaphragm under isoflurane anesthesia, after which organs were harvested.

## Measurement of m⁶A Level

Total RNA was isolated using RNeasy Kit (Qiagen, Valencia, CA, USA) from heart tissue or cells according to the protocol of the manufacturer. The mRNA was then further purified using the Dynabeads mRNA Purification Kit (ThermoFisher). The m⁶A levels in mRNA were detected by colorimetric ELISA assay with the EpiQuik m⁶A RNA Methylation Quantification kit (Epigentek, P-9005). Measurements were carried out in triplicate for each sample and five hearts were used for each group.

## Echocardiography

After AAV9 virus injection, animals with or without heart injury were subjected to echocardiography analysis at indicated time points. Cardiac function was evaluated using the Vevo 2100 ultrasound system (Visualsonics, Toronto, Canada) equipped with a high-frequency (30 MHz) linear array transducer, as described previously (*Wu et al., 2021*).

## Histology

Mice were sacrificed and weighed to obtain total body weight (BW) at indicated time points. The heart was then harvested and weighed to obtain heart weight (HW) and HW/BW ratio. Harvested hearts were fixed in 4% paraformaldehyde (PFA)/PBS solution overnight at room temperature, dehydrated in an ethanol series, and then processed for paraffin embedding. Paraffin sections were cut in 5 µm thickness. Sections were subjected to Masson's trichrome staining according to standard procedures. Fibrotic scar size was measured using the CaseViewer version 2.1 software (3DHISTECH, Hungary). The whole images of hearts were captured by Leica M205FA stereo fluorescence microscope to visualize the apex regeneration at indicated time points.

## Immunofluorescence staining

Hearts were embedded in paraffin and cut in 5 µm sections, deparaffinized, rehydrated and antigen retrieval. Sections were permeabilized with 0.5% Triton X-100/PBS and then blocked with 5% goat serum (Jackson ImmunoResearch Laboratories, USA) for 1 hr at room temperature, and incubated with primary antibodies overnight at 4℃. After washing with PBS, sections were incubated with corresponding secondary antibodies conjugated to fluorescence for 1 hr at room temperature, followed by counterstaining with DAPI (Sigma). Primary antibodies are as follows: anti-Mettl3 (Abcam,

ab195352, 1:500), anti-m[6]A (Synaptic Systems, 202111, 1:100), anti-Ki67 (Abcam, ab16667, 1:250), anti-phospho-Histone H3 (pH3) (CST, #53348 S, 1:400), anti-aurora kinase B (AurkB) (Abcam, ab2254, 1:200,) anti-GFP (Proteintech, 50430–2-AP, 1:200), and anti-Cardiac Troponin T (cTnT) (ThermoFisher, MA512960, 1:200). Secondary antibodies used are following: Alexa Fluor 488 goat anti-mouse or anti-rabbit IgG (Jackson ImmunoLabs, 115-545-071 or 111-545-003, 1:200), and Cy3-conjugated Affinipure Goat anti-mouse or anti-rabbit IgG (Proteintech, SA00009-1 or SA00009-2, 1:300). The slides were imaged with fluorescence microscope (Leica Microsystems) or Zeiss LSM 700 laser confocal microscope (Carl Zeiss).

## Wheat germ agglutinin (WGA) staining

Following deparaffinized, rehydrated, slides were then incubated with WGA conjugated to Alexa Fluor 488 (Invitrogen, W11261, 5 µg/ml) for 10 min at room temperature. To quantify the cell size, 5 independent hearts per group (at least 300 cells) were captured near apex with laser-scanning confocal microscope (LSM 700, Zeiss). ZEN 2012 lite software (Zeiss) was used to quantify the size of each cell.

## EdU labeling assay in vivo

For EdU labeling experiments, neonatal mice were subcutaneously injected with 50 µl of a 2 mg/ml solution of EdU (RiboBio, Guangzhou, China) dissolved in sterile water. Hearts were embedded in Tissue-Tek optimal cutting temperature compound (OCT) (Sakura, USA) for frozen section (4 µm). Sections were rinsed three times in PBS and fixed in 4% parapormaldehyde for 30 min. After rinsing three times again, citrate antigen retrieval was performed as described above. Sections were then incubated with 2 mg/mL glycine solution for 10 min, permeabilized with 0.5% Triton X-100 in PBS for 10 min, and then rinsed with PBS once for 5 min. This was followed by incubation with Apollo576 staining solution (1×) at room temperature for 30 min. Permeabilization was performed again with 0.5% Triton X-100 in PBS twice for 10 min. Sections were then rinsed with methanol for 5 min, washed with PBS once for 5 min, blocked with 5% goat serum for 1 h, and followed by incubation with primary antibody against cTnT (ThermoFisher, MA512960, 1:200) overnight. The following day, incubation with anti-mouse secondary antibody conjugated to Alexa Fluor 488 (1:200 dilution, Jackson ImmunoResearch Laboratories, USA) was performed. Sections were washed three times in PBS, stained with DAPI for 10 min to label nuclei, and mounted in Antifade Mounting Medium. Images were captured by laser-scanning confocal microscope (LSM 700, Zeiss) and analyzed by ZEN 2012 software (Zeiss).

To analyze CMs proliferation at the indicated time points, EdU was injected 8 hr prior to heart collection. For EdU pulse-chase experiments, EdU was injected once every 2 days to label all proliferating CMs during the whole period of cardiac regeneration. The last injection was performed 8 hr prior to heart collection. Sham-operated mice underwent the same procedure without the apical resection.

## RNA-seq

For total RNA isolation, neonatal mice were subjected to cardiac apical resection at postnatal day 3 (p3). Hearts were then extracted in sham and injured neonatal mice at 5 days post-resection (dpr), respectively. Three ventricles per group were used for RNA-sequencing analysis. RNA preparation, library construction, and sequencing on GBISEQ-500 platform were performed as previously described (*Xin et al., 2017*). After filtering the reads with low quality, clean reads were then obtained and mapped to the reference genome of mouse (GRCm38.p6) with HISAT (*Kim et al., 2015*). Genes expression level was quantified by a software package called RSEM (*Li and Dewey, 2011*) and expressed as fragments per kilobase of exon per million fragments mapped (FPKM). Differentially expressed (DE) genes were detected using NOISeq method (*Tarazona et al., 2011*) with Probability ≥0.8 and fold change (FC) of FPKM ≥2. Only those genes were considered for the differential expression analysis, which displayed FPKM ≥1 in either of the two samples under comparison. GO analysis was performed using online tool DAVID 6.8 (https://david.ncifcrf.gov/summary.jsp), and terms with *P*-value ≤0.05 were included. Differentially expressed gene heat maps were clustered by hierarchical clustering using cluster software (*Eisen et al., 1998*). Gene set enrichment analysis (GSEA) was performed to identify gene sets from signaling pathways that showed statistical differences between two groups by using GSEA software (http://software.broadinstitute.org/gsea/index.jsp).

## MeRIP-seq

Total RNAs isolated from neonatal hearts at 5 dpr and sham-operated hearts were subjected to MeRIP-seq. Eight neonatal hearts were pooled with at least two biological duplicates for each group. MeRIP-Seq was performed using 200 µg total RNA by Cloudseq Biotech Inc (Shanghai, China). Briefly, fragmented RNA was incubated with anti-m⁶A polyclonal antibody (Synaptic Systems, 202003) in IPP buffer for 2 hr at 4 °C. The mixture was then immunoprecipitated by incubation with protein-A beads (Thermo Fisher) at 4 °C for an additional 2 hr. Then, bound RNA was eluted from the beads with N6-methyladenosine (BERRY & ASSOCIATES, PR3732) in IPP buffer and then extracted with Trizol reagent (Thermo Fisher) by following the manufacturer's instruction. Purified RNA was used for RNA-seq library generation with NEBNext Ultra RNA Library Prep Kit (NEB). Both the input sample without immunoprecipitation and the m⁶A IP samples were subjected to 150 bp paired-end sequencing on Illumina HiSeq sequencer. For data analysis, Paired-end reads were harvested from Illumina HiSeq 4000 sequencer, and were quality controlled by Q30. After 3' adaptor-trimming and the lower quality reads removing by cutadapt software (v1.9.3). First, clean reads of all libraries were aligned to the reference genome (UCSC MM10) by Hisat2 software (v2.0.4). Methylated sites on RNAs (peaks) were identified by MACS software. Differentially methylated sites were identified by diffReps. These peaks identified by both software overlapping with exons of mRNA were figured out and chosen by home-made scripts. Differential peak analyses of MeRIP-seq data sets were performed by using a modification of the exomePeak R/Bioconductor package to compare the ratio of the absolute number of MeRIP reads with nonimmunoprecipitation reads at a given peak between 2 conditions (*Meng et al., 2014*; *Mathiyalagan et al., 2019*). Both RNA-seq and MeRIP-seq can be accessed in the Sequence Read Archive (SRA) under accession numbers SRP224051.

## RNA extraction and quantitative real-time PCR (qPCR)

Total RNA was isolated using RNeasy Kit (Qiagen, Valencia, CA, USA) from cells or heart tissue according to the protocol of the manufacturer, respectively. Reverse transcription to cDNA was performed with 30 ng of total RNA, random primers, and SuperScript III Reverse Transcriptase (Roche, USA). The qPCR was performed using a Light Cycler 480 SYBR Green I Master (Roche, USA) and the MiniOpticon qPCR System (Bio-Rad, CA, USA). After denaturation for 10 min at 95 °C, the reactions were subjected to 45 cycles of 95 °C for 30 s, 60 °C for 30 s, and 72 °C for 30 s. GAPDH was used as the internal standard control to normalize gene expression using the $2^{-\Delta\Delta Ct}$ method. The sequences of the qPCR primers were listed in *Supplementary files 2 and 3*.

## Protein extracts and Western blotting

Tissues or cells for SDS-PAGE were lysed in RIPA buffer (Beyotime Biotechnology) containing protease inhibitors (Sigma). Protein concentration was determined using the Bio-Rad Protein Assay (Bio-Rad Laboratories). 30 µg of protein were separated by SDA-PAGE, proteins were transferred onto PVDF membranes (Millipore), then blocked in 5% nonfat milk/TBS-Tween 20 and incubated with primary antibodies (dilution in TBST) overnight at 4 °C. Membranes were then washed and incubated with corresponding second antibodies for 1 hr at room temperature. Bands were detected by chemiluminescence reagents (ThermoFisher Scientific). Primary antibodies used are following: anti-Mettl3 (Abcam, ab195352, 1:1000), anti-Mettl14 (Abcam, ab98166, 1:1000), anti-Fgf16 (Santa Cruz, sc-390547, 1:500), anti-GFP (CST, #2956, 1:1000), anti-Flag (CST, #14793, 1:1000), and anti-β-actin (Proteintech, 60008–1-Ig, 1:2000). Secondary antibodies used are following: goat-anti-mouse horseradish peroxidase (HRP)-conjugated antibody (CST, #7076, 1:3000) and goat-anti-rabbit horseradish peroxidase (HRP)-conjugated antibody (CST, #7074, 1:2000). Quantitation of the chemiluminescent signal was analyzed using Image-Pro Plus version 6.0 (Media Cybernetics, Bethesda, MD). The relative expression levels of target protein/β-actin were set as one. All blots derive from the same experiment and were processed in parallel. The uncropped blots are listed in the Source data.

## Luciferase reporter assay

The pCMV-Gaussia-Dura Luc and pTK-Red Firefly Luc plasmids from ThermoFisher were used to construct the dual-luciferase reporter plasmid (pGL-RF), which contained both a Gaussia luciferase (GL) and a red firefly luciferase (RF). In brief, a new combined DNA fragment (Gaussia Dura Luc-SV40pA-pTK-Red Firefly Luc-bGHpA, GL-SV40pA-pTK-RF-bGHpA) was synthesized by Shanghai Generay

Biotech Co., Ltd (Shanghai, China). In the synthesized fragment, *XhoI* and *ClaI* restriction sites were inserted upstream and downstream of the termination codon of GL gene, respectively, to subclone Fgf16 coding sequences as below. The pTK-RF-bGHpA cassette from pTK-Red Firefly Luc plasmid was synthesized with replacement of *BamHI* restriction site between pTK promoter and RF gene with *XbaI*. The pCMV-Gaussia-Dura Luc plasmid was digested by *BamHI* and *NaeI* enzymes to remove the GL-bGHpA-SV40 promoter-Puromycin resistant gene (GL-bGHpA-SV40p-PuroR) fragment. The residual plasmid frame was then inserted by the synthesized combined fragment (GL-SV40pA-pTK-RF-bGHpA) using the EasyGeno Single Assembly Cloning kit (Tiangen Biotech, Beijing, China) to construct the dual-luciferase reporter plasmid (pGL-RF). The DNA sequence of pGL-RF plasmid can be found in *Supplementary file 4*. To constructed the wild-type Fgf16 reporter plasmid (pGL-Fgf16), the full-length of Fgf16 coding sequence (NM_030614.2) was amplified and subcloned into the pGL-RF plasmid linearized by digestion of *XhoI* and *ClaI* enzymes, thereby expressing the fusion protein of Gaussia-Fgf16. To make the mutant Fgf16 reporter plasmid (pGL-ΔFgf16), the adenosine bases in the m$^6$A consensus sequence (GGACT) were replaced with guanine (GGGCT) using a KOD-Plus-Mutagenesis Kit (Toyobo). H9c2 cells seeded in 96 well plates were co-transfected with reporter plasmid and pcDNA3-Mettl3 (Addgene, #53739) (100 ng for each) using LipoFiter Liposomal Transfection Reagent (Hanbio Biotechnology). An Mettl3 catalytic mutant (AA395-398, DPPW to APPA) was generated using a KOD-Plus-Mutagenesis Kit (Toyobo) to generate pcDNA-ΔMettl3 as described previously (*Li et al., 2019*). Two days after transfection, luciferase activity was examined using the Pierce Gaussia-Firefly Luciferase Dual Assay Kit (ThermoFisher Scientific) according to the manufacturer's protocol. Relative luciferase activity was measured using a BioTek Synergy 4 multimode microplate reader (BioTek Instruments). The activity of the Gaussia luciferase was normalized with that of Firefly luciferase. In addition, the mRNA expression levels of Gaussia luciferase were examined by qPCR as described above. Primers for Gaussia luciferase are as follows: Fw, 5'-ACC ACG GAT CTC GAT GCT GAC-3'; Re, 5'-ACT CTT TGT CGC CTT CGT AGG TG-3'. GAPDH was used as the internal standard control to normalize gene expression using the $2^{-\Delta\Delta Ct}$ method.

## MeRIP-qPCR and Ythdf2-RIP-qPCR

The m$^6$A-RIP-qPCR and Ythdf2-RIP-qPCR were performed according to a protocol as described previously (*Wang et al., 2018b*) with some modification. In brief, total RNA was isolated from neonatal hearts injected with AAV9-Mettl3 virus. The mRNA was then further purified using the Dynabeads mRNA Purification Kit (ThermoFisher) and fragmented by the RNA Fragmentation Reagents (ThermoFisher). The fragmented mRNA was incubated with anti-m$^6$A antibody (Synaptic Systems), anti-Ythdf2 antibody (Proteintech), or IgG in IPP buffer (150 mM NaCl, 10 mM TRIS-HCL and 0.1% NP-40) for 4 hr, followed by incubation with the eluted and blocked Dynabeads Protein A (ThermoFisher) for 2 hr at 4 °C. The mRNA binding to beads was eluted and purified with RNeasy MinElute Spin columns (Qiagen), followed by cDNA synthesis. qPCR was then performed as above described. GAPDH was used the control. The m$^6$A or Ythdf2 enrichment of Fgf16 was evaluated by $2^{-\Delta\Delta Ct}$ method. Primers used for RIP-qPCR in this study are as follows: RIP-Fw, 5'-GAC CAC AGC CGC TTC GGA AT-3'; RIP-Re, 5'-CGA TCC ATA GAG CTC TCC TCG C-3'.

## Measurement of Fgf16 mRNA stability

Primary cardiomyocytes were isolated from neonatal mice and treated with vehicle or Actinomycin-D (Sigma-Aldrich) at a concentration of 5 μM for 0, 3, and 6 hr, respectively. RNA was extracted at the indicated timepoints. qPCR analysis was then performed as described in 'RNA extraction and quantitative Real-Time PCR (qPCR)' section. Actinomycin-D was used to inhibit global mRNA transcription, and the ratio of *Fgf16* mRNA in Actinomycin-D-treated cells relative to vehicle-treated cells (A/V ratio) was calculated to evaluate the stability of *Fgf16* mRNA.

## Statistics

All statistics were calculated using GraphPad Prism 8 Software. Among three or more groups, statistical analysis was performed using one-way or two-way ANOVA with Dunnett's multiple comparisons post hoc tests. Comparisons between two groups were analyzed using unpaired and 2-tailed Student's t-test. All data are presented as the mean ± SEM. A p value of less than 0.05 was considered statistically significant.

## Acknowledgements

This work was supported by grants from the National Key R&D Program of China (2017YFA0103302), the National Natural Science Foundation of China (82070257, 81770240, 81570222, 81670422, 81873517), the Guangdong Natural Science Funds for Distinguished Young Scholar (2014A030306011), the Guangdong Science and Technology Planning Project (2014A050503043), the New Star of Pearl River on Science and Technology of Guangzhou (2014J2200002), the Top Young Talents of Guangdong Province Special Support Program (87315007), the Young Taishan Scholars Program of Shandong Province (tsqn20161045), and the Research Grant of Key Laboratory of Regenerative Medicine, Ministry of Education, Jinan University (ZSYXM202004, ZSYXM202104, and ZSYXM202206), China.

## Additional information

### Funding

| Funder | Grant reference number | Author |
| --- | --- | --- |
| Ministry of Science and Technology of the People's Republic of China | 2017YFA0103302 | Zhenyu Ju |
| National Natural Science Foundation of China | 82070257 | Xu-Feng Qi |
| National Natural Science Foundation of China | 81770240 | Xu-Feng Qi |
| National Natural Science Foundation of China | 81570222 | Xu-Feng Qi |
| National Natural Science Foundation of China | 81670422 and 81873517 | Guo-Hua Song |
| Department of Science and Technology of Guangdong Province | 2014A030306011 | Xu-Feng Qi |
| Department of Science and Technology of Guangdong Province | 2014A050503043 | Xu-Feng Qi |
| Department of Science and Technology of Guangdong Province | 87315007 | Xu-Feng Qi |
| Guangzhou Municipal Science and Technology Bureau | 2014J2200002 | Xu-Feng Qi |
| Department of Science and Technology of Shandong Province | tsqn20161045 | Guo-Hua Song |
| Jinan University | ZSYXM202004 | Xu-Feng Qi |
| Jinan University | ZSYXM202104 | Xu-Feng Qi |
| Jinan University | ZSYXM202206 | Xu-Feng Qi |

The funders had no role in study design, data collection and interpretation, or the decision to submit the work for publication.

### Author contributions

Fu-Qing Jiang, Kun Liu, Jia-Xuan Chen, Yan Cao, Wu-Yun Chen, Wan-Ling Zhao, Investigation; Guo-Hua Song, provided valuable comments and suggestions; Chi-Qian Liang, Methodology; Yi-Min Zhou, Software; Huan-Lei Huang, provided valuable comments and suggestions; Rui-Jin Huang, provided valuable comments and suggestions; Hui Zhao, provided valuable comments and

suggestions; Kyu-Sang Park, provided valuable comments and suggestions; Zhenyu Ju, Dongqing Cai, Writing – review and editing; Xu-Feng Qi, Supervision, Writing – original draft, Writing – review and editing

### Author ORCIDs
Kun Liu  http://orcid.org/0000-0003-3512-4516
Xu-Feng Qi  http://orcid.org/0000-0002-5911-071X

### Ethics
All animal studies were approved by the Institutional Animal Care and Use Committee of Jinan University (IACUC-2018921-03).

### Decision letter and Author response
Decision letter https://doi.org/10.7554/eLife.77014.sa1
Author response https://doi.org/10.7554/eLife.77014.sa2

## Additional files

### Supplementary files
- Transparent reporting form
- Supplementary file 1. Sequences of siRNAs used in this study.
- Supplementary file 2. Primer sequences for real-time PCR analysis in mouse.
- Supplementary file 3. Primer sequences for real-time PCR analysis in H9c2 cells.
- Supplementary file 4. Full plasmid sequence of the pGL-RF vector.

### Data availability
Both RNA-seq and MeRIP-seq can be accessed in the Sequence Read Archive (SRA) under accession numbers SRP224051.

The following previously published dataset was used:

| Author(s) | Year | Dataset title | Dataset URL | Database and Identifier |
|---|---|---|---|---|
| Jinan University | 2020 | Mettl3-dependent m6A modification regulates heart regeneration by modulating Fgf16 expression in mice | https://www.ncbi.nlm.nih.gov/sra/?term=SRP224051 | NCBI Sequence Read Archive, SRP224051 |

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
