## [Editor Report]

This study explores a novel mechanistic insight into the epitranscriptomic regulation of murine heart regeneration. The authors have demonstrated that Mettl3 post-transcriptionally reduces Fgf16 mRNA levels through an m6 A-Ythdf2-dependent pathway, thereby restricting cardiomyocyte proliferation and heart regeneration in postnatal mice. Targeting the Mettl3/m6A/Ythdf2/Fgf16 pathway may represent a promising therapeutic strategy to promote the proliferation of cardiomyocytes in mammals.

---

## [Decision Letter]

**Decision letter after peer review:**

Thank you for submitting your article "Mettl3-mediated m^6^A modification of Fgf16 restricts cardiomyocyte proliferation during heart regeneration" for consideration by *eLife*. Your article has been reviewed by 2 peer reviewers, and the evaluation has been overseen by a Reviewing Editor and Balram Bhargava as the Senior Editor. The following individuals involved in the review of your submission have agreed to reveal their identity: Nan Wang (Reviewer #1); Prabhu Mathiyalagan (Reviewer #2).

Essential revisions:

This study has uncovered a novel mechanism in the regulation of cardiomyocyte proliferation and heart regeneration following heart injury via the regulation of RNA methylation. Overall, the data and results support the major claims of the paper, particularly in neonatal models. However, some of the claims appear overstated with additional evidence needed to support the authors' claims.

1. In neonatal models, Mettl3 expression was markedly increased following heart injury. It is not clear whether Mettl3 expression was altered in postnatal and, particularly, in adult mouse heart injury models. While down-regulation of Mettl3 expression in postnatal heart injury models also increased myocardial proliferation as assessed by Ki67+ or pH3+ CMs, the overall Ki67+ or pH3+ CMs were almost 10 fold lower than in the neonatal heart injury models (Figure 3I and 3N vs Figure 6E, 6G), yet the CM cell size changes were as prevalent and dramatic as in the neonatal models. It is not clear how to explain these discrepancies and raises the question as to whether the improved tissue repair and cardiac function are primarily dependent on increased cardiomyocyte proliferation in this setting. A similar relatively low Ki67+ CMs were also shown in adult MI mouse models.

2. Fgf16 mRNA levels were increased by 2-fold when Mettl3 expression was down-regulated in the neonatal heart injury model (Figure 8I). While the authors claimed "mRNA level of Fgf16 in the neonatal heart was slightly elevated by AAV9-Fgf16 virus", the actual figure showed a dramatic 5-fold increase (Figure 9C). Yet, this dramatic 5-fold increase in Fgf16 mRNA did not significantly increase cardiomyocyte proliferation and heart regeneration or improve cardiac function. Increased CM proliferation and improved cardiac function were detected only with 10 fold increase in Fgf16 mRNA levels in mutant Fgf16 virus models. Together, it indicates that the impact of Mettl3 downregulation on CM proliferation, heart regeneration, and improved cardiac function cannot be explained by the 2-fold increase in Fgf16 mRNA levels.

3. The authors need to provide additional evidence to support the role of Fgf16 in Mettl3-mediated regulation of CM proliferation and heart regeneration. For instance, the authors may assess the impact of the sham control, Mettl3 shRNA, mutant Fgf16, and Mettl3 shRNA/mutant Fgf16. If the impact of Mettl3 down-regulation was mediated by up-regulation of Fgf16, no additive impact on CM proliferation, heart regeneration, and improved cardiac function would be expected when Mettl3 shRNA and mutant Fgf16 are co-expressed.

4. The manuscript reports 333 genes that show overlap between MeRIP-seq (i.e. increased m6A modification) and RNA-seq (i.e. upregulated for expression) at 5 dpr. On the other hand, at 5 dpr, the authors also show increased Mettl3 and m6A levels in the heart. How do the authors explain the contradiction that there seems to be a positive correlation between Mettl3 expression, m6A levels, and Fgf16 expression, while the proposed mechanism involves Mettl3-dependent degradation of Fgf16 mRNA as a potential mediator for decreased cardiac regeneration in these hearts? Should not be there a negative correlation between m6A-modified mRNAs and their expression? A detailed discussion on providing the rationale would improve readership.

5. Since Fgf16 mRNA expression is induced during heart regeneration, how specific are the reads that are mapped to FgF16 using MeRIP-seq are true reads mapping to m6A modification and not reads captured due to increased abundance for Fgf16 mRNA. This could be clarified by providing both RNA-seq and MeRIP-seq reads together in an IGV plot across Fgf16 gene (between sham and apr) to show specificity and distinction for Fgf16 peaks between the two enrichment methods.

6. Since the Mettl3-deficiency leads to increased cardiomyocyte proliferation in both apical and remote areas within the heart, would it be assumed that Mettl3-mediated m6A modifications on mRNAs are also dynamically regulated in remote areas within the heart in response to injury? Having Mettl3 and m6A levels quantified in remote areas would provide deeper insights into pathways regulated between the apical and remote areas of the heart post-injury.

7. Having the level of m6A determined in the CM and nCM cell types of the heart, would also add significant context to Mettl3-specific roles in CM than in nCMs.

8. In the MeRIP-seq method, mentioning the amount of Total RNA and Poly(A)+ RNA that was used for MeRIP-seq would provide some context on using this method for neonatal mouse hearts, as the RNA required for MeRIP-seq are usually in large amounts than for other methods.

9. How would the authors rationalize or discuss that the observed changes to m6A-dependent effects on mRNAs such as for Fgf16 are indeed Mettl3-dependent. Would overexpression or knockdown of another m6A regulatory protein (such as Mettl14 or FTO) have a similar impact on these mRNAs and therefore on cardiac regeneration?

---

## [Author Response]

Essential revisions:This study has uncovered a novel mechanism in the regulation of cardiomyocyte proliferation and heart regeneration following heart injury via the regulation of RNA methylation. Overall, the data and results support the major claims of the paper, particularly in neonatal models. However, some of the claims appear overstated with additional evidence needed to support the authors' claims.1. In neonatal models, Mettl3 expression was markedly increased following heart injury. It is not clear whether Mettl3 expression was altered in postnatal and, particularly, in adult mouse heart injury models. While down-regulation of Mettl3 expression in postnatal heart injury models also increased myocardial proliferation as assessed by Ki67+ or pH3+ CMs, the overall Ki67+ or pH3+ CMs were almost 10 fold lower than in the neonatal heart injury models (Figure 3I and 3N vs Figure 6E, 6G), yet the CM cell size changes were as prevalent and dramatic as in the neonatal models. It is not clear how to explain these discrepancies and raises the question as to whether the improved tissue repair and cardiac function are primarily dependent on increased cardiomyocyte proliferation in this setting. A similar relatively low Ki67+ CMs were also shown in adult MI mouse models.

Thanks very much for the comments from reviewers!

According to the suggestion from reviewers, we further examined the expression of Mettl3 in postnatal and adult mice upon heart injury in the revised manuscript. Myocardial infarction (MI) injury was performed in postnatal (p7) and adult (3-month-old) mice, respectively. Heart tissues were then isolated at 5 day post injuries for Western blotting assay. In contrast with the data from neonatal mice model, our data showed that heart injury has no significant effects on the expression of Mettl3 in postnatal (p7) and adult mice (Figure 6—figure supplement 1). These findings suggest that Mettl3 has different responses to heart injuries in the regenerative stage compared with non-regenerative stage in mice. These different responses of Mettl3 to heart injury in regenerative and non-regenerative stages might results from the diverse microenvironments in neonatal and adult heats.

Regarding the effects of Mettl3 on cardiomyocyte proliferation, our immunofluorescent staining results showed that Mettl3 knockdown significantly promotes cardiomyocyte proliferation in neonatal and postnatal as well as adult mice compared with controls. Indeed, we found that cardiomyocyte proliferation levels in neonatal are higher than that in postnatal and adult mice regardless of Mettl3 knockdown. It is well known that neonatal mice have the capacity to regenerate the injured heart mainly through cardiomyocyte proliferation, but the regenerative capacity of heart is lost at postnatal day 7 with a dramatic decrease in the ability of cardiomyocyte proliferation (Porrello et al., Science, 2011, 331: 1078-1080; Porrello et al., PNAS, 2013, 110:187-192; Mahmoud et al., Nature, 2013, 497:249-253). These previous studies indicate that heart regenerative potential and cardiomyocyte proliferative ability are gradually lost during mouse heart development and maturation, especially in postnatal stage. In other words, the basic ability of cardiomyocyte proliferation is indeed lower in postnatal mice compared with neonatal mice. These previously demonstrated phenomena are consistent with the observation in our study showing a lower proliferation of cardiomyocytes in postnatal heart injury model compared with neonatal model, regardless of Mettl3 knockdown. Although the lower proliferation of cardiomyocytes in postnatal heart injury models, Mettl3 knockdown-induced increase ratio of cardiomyocyte proliferation compared with control is comparable between neonatal (Figure 3O) and postnatal (Figure 6E) heart injury models (both about 2~3 folds).

In addition, WGA staining revealed that cardiomyocyte size is smaller in Mettl3-silencing neonatal heart during regeneration compared control, implying the increased numbers of cardiomyocytes in Mettl3-deficient neonatal hearts (smaller cardiomyocyte size with no change in heart to body weight ratio as shown in Figure 3F). As expected, increased cardiomyocyte proliferation levels were indeed detected by immunofluorescent staining using antibodies of proliferating markers in Mettl3-deficient neonatal injury heart. Consistent with neonatal heart injury model, Mettl3 knockdown-induced smaller cardiomyocyte size was also observed in postnatal and adult heart injury models. These findings indicate that Mettl3 knockdown might prevent the potential cardiac hypertrophy during heart injury and regeneration or repair, in addition to promoting cardiomyocyte proliferation. In fact, it is known that heart injury such as myocardium infarction indeed results in cardiac hypertrophy especially in adult mice.

Taken these findings together, we suppose that Mettl3 knockdown improves heart repair and cardiac function upon injury, at least in part, through increasing cardiomyocyte proliferation and controlling cardiac hypertrophy. Of course, we can not exclude other potential influence which may contribute to Mettl3 knockdown-induced improvement of heart regeneration and cardiac repair.

We hope that these additional experiments and response to the comments will make our revised manuscript to be more suitable for publication in this journal.

2. Fgf16 mRNA levels were increased by 2-fold when Mettl3 expression was down-regulated in the neonatal heart injury model (Figure 8I). While the authors claimed "mRNA level of Fgf16 in the neonatal heart was slightly elevated by AAV9-Fgf16 virus", the actual figure showed a dramatic 5-fold increase (Figure 9C). Yet, this dramatic 5-fold increase in Fgf16 mRNA did not significantly increase cardiomyocyte proliferation and heart regeneration or improve cardiac function. Increased CM proliferation and improved cardiac function were detected only with 10 fold increase in Fgf16 mRNA levels in mutant Fgf16 virus models. Together, it indicates that the impact of Mettl3 downregulation on CM proliferation, heart regeneration, and improved cardiac function cannot be explained by the 2-fold increase in Fgf16 mRNA levels.

As mentioned by reviewers, our original data revealed that Fgf16 mRNA levels were increased by about 2-fold when Mettl3 expression was down-regulated in neonatal heart injury model (Figure 8I). However, the protein levels of Fgf16 were increased by about 8-fold when Mettl3 expression was down-regulated in the neonatal heart injury model (Figure 8K). It is well known that the biological function of Fgf16 mainly depends on its levels of protein rather than mRNA. Therefore, it is reasonable that the higher increase in Fgf16 protein levels induced by Mettl3 downregulation can promote cardiomyocyte proliferation, heart regeneration, and cardiac function in the neonatal heart injury model. We speculate that the different levels between Fgf16 mRNA and its protein maybe due to the dynamic process of protein translation. In the neonatal heart injury model, it is possible that Mettl3 downregulation reduces the m^6^A modification of the endogenous Fgf16 mRNAs and blocks Ythdf2-mediated mRNA decay, which in turn promotes the translation of Fgf16 proteins. That is why Fgf16 protein levels are higher than its mRNA levels when Mettl3 was down-regulated under our experimental conditions.

Regarding the overexpression of exogenous Fgf16, our original data showed that AAV9-Fgf16 injection increased Fgf16 mRNA by about 5-fold. However, this 5-fold increase in wild-type Fgf16 mRNA did not significantly increase cardiomyocyte proliferation and heart regeneration. As mentioned above, the biological function of Fgf16 mainly depends on its protein levels instead of mRNA levels. To further determine this possible, western blotting was further performed to detect Fgf16 protein levels in this revised manuscript. Our data revealed that AAV9-Fgf16 injection only slightly increased Fgf16 protein levels by about 2-fold. However, AAV9-ΔFgf16 injection greatly increased Fgf16 protein levels by about 9-fold. These findings are consistent with our data showing that overexpression of ΔFgf16, rather than wild-type Fgf16, significantly promoted cardiomyocyte proliferation and heart regeneration under our experimental conditions. We think that the different increase levels of wild-type Fgf16 mRNA and proteins maybe due to the dynamic process of mRNA decay and protein translation. AAV9-Fgf16 injection indeed dramatically increase the mRNA levels of wild-type Fgf16 (about 5-fold as qPCR validation). Heart injury significantly promoted Mettl3 expression and m^6^A modification (Figure 1F-I), which might accelerate the decay of exogenous Fgf16 mRNA overexpressed by AAV9-Fgf16 through Ythdf2. Consistent with this hypothesis, the mRNA levels of Fgf16 in AAV9-ΔFgf16 group was significantly higher than that in AAV9-Fgf16 group (about 10-fold versus 5-fold). Under our experimental conditions, AAV9-Fgf16 injection greatly increased Fgf16 mRNA levels (about 5-fold) but slightly elevated Fgf16 protein levels (about 2-fold), which suggest that higher m^6^A modification in the wild-type Fgf16 mRNA might restrict its translation into proteins in addition to accelerating its mRNA decay. Consistent with our findings, previous study has reported that increased m^6^A modification can indeed suppress the local translation of GAP-43 mRNA in axons (Yu et al., Nucleic Acids Res, 2018, 46(3):1412-1423).

3. The authors need to provide additional evidence to support the role of Fgf16 in Mettl3-mediated regulation of CM proliferation and heart regeneration. For instance, the authors may assess the impact of the sham control, Mettl3 shRNA, mutant Fgf16, and Mettl3 shRNA/mutant Fgf16. If the impact of Mettl3 down-regulation was mediated by up-regulation of Fgf16, no additive impact on CM proliferation, heart regeneration, and improved cardiac function would be expected when Mettl3 shRNA and mutant Fgf16 are co-expressed.

According to the suggestion from reviewers, we performed additional experiments to further support the role of Fgf16 in Mettl3-mediated regulation of CM proliferation and heart regeneration in this revised manuscript. However, it seems that the reviewer’s strategy is not enough to support the role of Fgf16 by using mutant Fgf16 together with Mettl3 shRNA. In our original manuscript, both in vitro and in vivo data strongly evidenced that Mettl3 deficiency greatly promotes CM proliferation and heart regeneration (Figures 2-3). Meanwhile, Mettl3 deficiency significantly increased Fgf16 protein levels in CMs and myocardium with and without heart injury (Figure 8J-M). More importantly, our original data have demonstrated that Fgf16 deficiency significantly suppressed CM proliferation, which indicating that Fgf16 plays an important role in CM proliferation (Figure 8A-D). These findings suggest that Fgf16 might be important for Mettl3-mediated regulation of CM proliferation and heart regeneration.

To further evidence the direct role of Fgf16 in Mettl3-mediated regulation of CM proliferation and heart regeneration, we further performed additional in vitro and in vivo experiments to assess the impact of Fgf16 knockdown on Mettl3-mediated regulation of CM proliferation and heart regeneration in this revised manuscript. Firstly, primary cardiomyocytes isolated from neonatal mice were treated with siRNA control (siNC), siMettl3, and siMettl3+siFgf16 for 48 h, respectively. Ki67 and cTnT double staining revealed that siMettl3 treatment greatly increased the percentage of Ki67^+^ cTnT^+^ cells compared with control group, indicating an increased proliferation level of CMs. However, siMettl3-increased CM proliferation was significantly blocked by siFgf16 treatment (Figure 8—figure supplement 6A and B). These in vitro data suggest that suppressing Fgf16 expression in Mettl3-deficient CMs can greatly block its proliferation. To further confirm this idea, we also injected neonatal mice at p1 by AAV9-shNC, AAV9-shMettl3, and AAV9-shMettl3+AAV9-shFgf16, respectively. Apex resection was then performed at p3. Heart tissues were collected at 5 dpr and 14 dpr for analysis, respectively (Figure 8—figure supplement 6C). In agreement with in vitro data, immunofluorescent staining revealed that shMettl3 injection increased CM proliferation in ventricular apex at 5 dpr compared with control groups. However, the shMettl3-increased CM proliferation was significantly blocked by shFgf16 injection (Figure 8—figure supplement 6D and E). In addition, histological analysis revealed that scar size in the cardiac apex at 14 dpr was greatly decreased by Mettl3 deficiency compared with control groups. But this shMettl3-induced decrease in scar size was significantly blocked by Fgf16 knockdown (Figure 8—figure supplement 6F and G). In consistent, cardiac function analysis revealed that shMettl3-inceased left ventricular systolic function was significantly blocked by shFgf16 (Figure 8—figure supplement 6H-J). Taken together, our data strongly evidenced the important role of Fgf16 in Mettl3-mediated regulation of CM proliferation and heart regeneration.

We hope that these additional experiments and response to the comments will make our revised manuscript to be more suitable for publication in this journal.

4. The manuscript reports 333 genes that show overlap between MeRIP-seq (i.e. increased m6A modification) and RNA-seq (i.e. upregulated for expression) at 5 dpr. On the other hand, at 5 dpr, the authors also show increased Mettl3 and m6A levels in the heart. How do the authors explain the contradiction that there seems to be a positive correlation between Mettl3 expression, m6A levels, and Fgf16 expression, while the proposed mechanism involves Mettl3-dependent degradation of Fgf16 mRNA as a potential mediator for decreased cardiac regeneration in these hearts? Should not be there a negative correlation between m6A-modified mRNAs and their expression? A detailed discussion on providing the rationale would improve readership.

According to the suggestion from reviewers, we further modified the discussion in the revised manuscript. Indeed, our data revealed that the levels of Mettl3, m6A, and Fgf16 were increased at 5 dpr compared with sham-operated hearts. It seems there is a positive correlation between Mettl3-m6A signaling and Fgf16 levels. To further determine the correlation, we examined Fgf16 expression in the injured heart with Mettl3 overexpression or silencing at 5 dpr. Our data showed that Mettl3 overexpression increased m6A enrichment in Fgf16 mRNA but suppressed Fgf16 expression levels. In contrast, Mettl3 deficiency decreased m6A enrichment in Fgf16 mRNA but promoted Fgf16 express levels (Figure 8G-M). These findings suggest that Mettl3-mediated m6A negatively regulates the Fgf16 expression during heart injury. Moreover, Mettl3 deficiency-increased cardiomyocyte proliferation and heart regeneration could be significantly blocked by Fgf16 silencing during heart regeneration (Figure 8—figure supplement 6), indicating the important role of Fgf16 in Mettl3-mediated regulation of heart regeneration. Given that the cardiac regenerative capacity of neonatal mouse only appears within one week after birth, we speculate that Fgf16 expression in neonatal heart is increased upon injury, thereby promoting heart regeneration. Meanwhile, the increased Mettl3 expression and m6A modification in the injured neonatal heart may be responsible for preventing excessive expression of Fgf16 through Ythdf2-mediated RNA decay, thereby balancing cardiac regenerative capacity. These responses were also discussed in our revised manuscript (5^th^ paragraph in the Discussion).

5. Since Fgf16 mRNA expression is induced during heart regeneration, how specific are the reads that are mapped to FgF16 using MeRIP-seq are true reads mapping to m6A modification and not reads captured due to increased abundance for Fgf16 mRNA. This could be clarified by providing both RNA-seq and MeRIP-seq reads together in an IGV plot across Fgf16 gene (between sham and apr) to show specificity and distinction for Fgf16 peaks between the two enrichment methods.

According to the suggestion from reviewers, we provided both RNA-seq and MeRIP-seq reads together in IGV plot across Fgf16 gene in this revised manuscript. MeRIP-seq showed a great increase (about 10 folds) in the m6A peaks of Fgf16 mRNA during neonatal heart injury (Figure 8F, upper panel). However, the RNA-seq reads were only increased by 2 folds in the Fgf16 mRNA during neonatal heart injury (Figure 8F, lower panel). These disproportionate increases indicate that the MeRIP-seq reads are true reads mapping to m6A modification and not reads captured due to increased abundance for Fgf16 mRNA. These responses were also discussed in our revised manuscript (2^nd^-4^th^ sentences in 2^nd^ paragraph in the Result section “Mettl3-mediated m^6^A modification downregulates Fgf16 expression in cardiomyocytes”).

6. Since the Mettl3-deficiency leads to increased cardiomyocyte proliferation in both apical and remote areas within the heart, would it be assumed that Mettl3-mediated m6A modifications on mRNAs are also dynamically regulated in remote areas within the heart in response to injury? Having Mettl3 and m6A levels quantified in remote areas would provide deeper insights into pathways regulated between the apical and remote areas of the heart post-injury.

According to the suggestion from reviewers, we performed additional experiments in the revised manuscript to determine Mettl3 and m6A levels in the remote areas during heart injury. Our original data showed that cardiac injury increased Mettl3 expression and m6A levels in apical areas at 5 dpr (Figure 1F-J). To further determine the response of Mettl3 expression and m6A modification in remote areas to apical resection injury, we collected myocardium tissues from remote areas and performed Western blotting and ELISA assays, respectively. In consistent with apical areas, our data showed that Mettl3 expression and m6A levels were also increased in remote areas at 5 dpr compared with control groups (Figure 1—figure supplement 1). These data suggest that heart injury could induce the same regulation of Mettl3-m6A signaling in both apical and remote areas. This idea was further supported by our original data showing that Mettl3 deficiency promoted cardiomyocyte proliferation in both apical and remote areas at 5 dpr (Figure 3 and Figure 3—figure supplement 2).

7. Having the level of m6A determined in the CM and nCM cell types of the heart, would also add significant context to Mettl3-specific roles in CM than in nCMs.

According to the suggestion from reviewers, we further determined the m6A levels in CM and nCM cell types in the neonatal heart. Primary CM and nCM cells were isolated from neonatal heart at p1, followed by RNA extraction and m6A level determination using colorimetric ELISA assay. We found that the levels of m6A in CMs is higher than that in nCMs, indicating the functional importance of m6A modification in CMs in neonatal mice (Figure 1—figure supplement 2A). To determine the response of Mettl3 expression in CMs and nCMs to injury, primary CMs and nCMs were isolated from sham and injured neonatal hearts at 5 dpr. Our data showed that Mettl3 expression was significantly increased in CMs rather than nCMs upon injury (Figure 1—figure supplement 2B). In addition, AAV9 system was used to inhibit or overexpress Mettl3 expression specifically in CMs in this study. Taken together, these additional experiments further support the functional importance of Mettl3-m6A signaling in CMs during heart injury and regeneration.

8. In the MeRIP-seq method, mentioning the amount of Total RNA and Poly(A)+ RNA that was used for MeRIP-seq would provide some context on using this method for neonatal mouse hearts, as the RNA required for MeRIP-seq are usually in large amounts than for other methods.

According to the suggestion from reviewers, we modified the description of MeRIP-seq method in our revised manuscript as follows: “Total RNAs isolated from neonatal hearts at 5 dpr and sham-operated hearts were subjected to MeRIP-seq. Eight neonatal hearts were pooled with at least two biological duplicates for each group. MeRIP-Seq was performed using 200 μg of total RNA by Cloudseq Biotech Inc (Shanghai, China). Briefly, fragmented RNA was……” (Please see the MeRIP-seq section in the “Materials and methods” part)

9. How would the authors rationalize or discuss that the observed changes to m6A-dependent effects on mRNAs such as for Fgf16 are indeed Mettl3-dependent. Would overexpression or knockdown of another m6A regulatory protein (such as Mettl14 or FTO) have a similar impact on these mRNAs and therefore on cardiac regeneration?

In the present study, four popular regulators (Mettl3, Mettl14, Alkbh5, and Fto) of m6A modification were determined during neonatal heart growth and regeneration. Among these four regulators, Mettl3 rather than other three ones were significantly increased in mRNA and protein levels at p14 (non-regenerative stage) compared p3 (regenerative stage) (Figure 1A-C). In consistent with these data, apical resection injury significantly increased mRNA and protein levels of Mettl3 rather than other three regulators compared with sham groups (Figure 1G-J). In agreement with Mettl3 change, cardiac levels of m6A modification were indeed significantly increased during neonatal heart growth (Figure 1D) and apical resection injury (Figure 1F). These data indicate that Mettl3-mediated m6A modification might play an important role in neonatal heart regeneration.

To further determine whether and how m^6^A modification regulates *Fgf16* mRNA levels, we performed m^6^A-RIP-qPCR and found that m^6^A enrichment in *Fgf16* was remarkably upregulated by *Mettl3* overexpression (Figure 8G) but suppressed by *Mettl3* silencing (Figure 8H) in the regenerating heart at 5 dpr. Fgf16 mRNA expression at 5 dpr was significantly decreased and increased by overexpressing and silencing Mettl3, respectively (Figure 8I). Furthermore, western blotting showed that Mettl3 knockdown (Figure 8J) significantly increased the production of Fgf16 protein in the regenerating heart at 5 dpr (Figure 8K). Taken together, these findings indicate that Mettl3-mediated m^6^A negatively regulates the post-transcriptional levels of *Fgf16* during heart injury.

According to the suggestion from reviewers, we further determined the potential effects of Mettl14 and Fto on Fgf16 expression and cardiomyocyte proliferation. Primary cardiomyocytes isolated from neonatal mice at p1 were treated with siMettl14 and siFto for 48 hours, followed by qPCR assay and immunofluorescent staining assay. Our data showed that Mettl14 and Fto knockdown (Figure 8—figure supplement 7A and B) did not significantly change the expression level of Fgf16 in primary cardiomyocytes (Figure 8—figure supplement 7C). Moreover, the percentage of Ki67-positive cardiomyocytes in siMettl14- and siFto-treated groups was comparable with control group (Figure 8—figure supplement 7D and E). These data suggest that Mettl14 and Fto knockdown has no significant effects on Fgf16 expression and cardiomyocyte proliferation.

Taken these findings together with our original data, it is reasonable that m6A-mediated Fgf16 mRNA modification, cardiomyocyte proliferation, and heart regeneration depend on Mettl3 rather than other m6A regulators.